



# Main drivers of transparent exopolymer particle distribution across the surface Atlantic Ocean

Marina Zamanillo[1], Eva Ortega–Retuerta[1,2], Sdena Nunes[1], Pablo Rodríguez–Ros[1], Marta Estrada[1], María Montserrat Sala[1], Rafel Simó[1]

[1]Biologia Marina i Oceanografia, Institut de Ciències del Mar, Consejo Superior de Investigaciones Científicas, Spain
[2]CNRS, Sorbonne Université, UMR 7621, Laboratoire d'Océanographie Microbienne, Banyuls–sur–Mer, France

*Correspondence to*: Rafel Simó (rsimo@icm.csic.es)

**Abstract.** Transparent exopolymer particles (TEP) are a class of gel particles produced mainly by microorganisms. TEP play an important role in the ocean carbon cycle, affect sea–air gas exchange and contribute to organic aerosols. The first step to evaluate the TEP influence in these processes is the prediction of TEP occurrence in the ocean. Yet, little is known about the physical and biological variables that control their abundance, particularly in the open ocean. Here we describe horizontal TEP distribution in the surface waters along a North–South transect in the Atlantic Ocean during October–November 2014. Physical and biological variables were run in parallel. Two main regions were separated due to remarkable differences; the open Atlantic Ocean (OAO, n = 30), and the Southwestern Atlantic Shelf (SWAS, n = 10). TEP concentration in the entire transect ranged from 18.3 to 446.8 μg XG eq L$^{-1}$ and averaged 117.1 ± 119.8 μg XG eq L$^{-1}$, with the maximum concentrations in the edge of the Canary Coastal Upwelling (CU, n = 1) and the SWAS, but with the highest TEP to chlorophyll *a* (TEP:Chl *a*) ratios at the OAO (CU excluded, average 183 ± 56) and CU (1760.4). TEP were significantly and positively related to Chl *a* and phytoplankton biomass, expressed in terms of C, along the entire transect. In the OAO, TEP were positively related to some phytoplankton groups, mainly to *Synechococcus*, and negatively related to the previous 24–hours averaged solar radiation, suggesting the predominance of TEP breaking above the induction of TEP production by UV radiation. Multiple regression analyses showed the combined positive effect of phytoplankton and heterotrophic prokaryotes (HP) on TEP distribution in this region. In the SWAS, TEP were positively related to high nucleic acid prokaryotic cells (HNA) and total phytoplankton biomass, but not with any particular phytoplankton group. TEP constituted an important portion of the particulate organic carbon (POC) pool in the entire transect (28.1–109.8 %), and was generally higher than the phytoplankton and HP fraction, highlighting the importance of TEP in the cycle of organic matter in the ocean.



## 1 Introduction

Transparent exopolymer particles (TEP) are defined as a class of non–living organic particles, mainly formed by acidic polysaccharides, which are stainable with Alcian Blue (Alldredge et al., 1993). Due to their stickiness, TEP favour the formation of large aggregates of organic matter, enhancing particle sinking in the ocean (Logan et al., 1995; Passow et al., 2001; Burd and Jackson, 2009). By contrast, due to their low density, TEP and TEP–rich microaggregates can also ascend in the water column and accumulate in the sea surface microlayer (Azetsu-Scott and Passow, 2004; Wurl et al., 2009) where they affect sea–air gas exchange (Calleja et al., 2008) or can be released to the atmosphere by bubble bursting (Zhou et al., 1998; Aller et al., 2005; Kuznetsova et al., 2005), contributing to organic aerosol and acting as cloud condensation nuclei and ice nucleating particles (Orellana et al., 2011; Leck et al., 2013; Wilson et al., 2015). The presence of TEP also affects the microbial food–web, as they can be used as a food source for zooplankton (Decho and Moriarty, 1990; Dilling et al., 1998; Ling and Alldredge, 2003) and heterotrophic prokaryotes (HP) (Passow, 2002b). TEP also provide surfaces for microbial colonization (Alldredge et al., 1986; Grossart et al., 2006; Azam and Malfatti, 2007).

TEP distribution in marine systems depends on the complex balance between the sources and the sinks (Alldredge et al., 1998; Passow, 2002a). TEP sinks include some of the above mentioned processes (sinking to the deep ocean, release to the atmosphere, grazing and degradation), and also photolysis by UV radiation (Ortega-Retuerta et al., 2009b). Regarding the sources, TEP are released mainly by microorganisms, during production and decomposition processes, either directly as detritus (Hong et al., 1997; Berman-Frank et al., 2007), or indirectly through the abiotic self–assembly of released precursors (Passow and Alldredge, 1994; Thuy et al., 2015). Phytoplankton are major TEP producers in the ocean, although HP are also able to produce TEP (Biddanda, 1986; Stoderegger and Herndl, 1998; Passow, 2002b; Ortega-Retuerta et al., 2010). Some phytoplankton groups that have been shown to produce TEP include cyanobacteria (Grossart et al., 1998; Mazuecos, 2015; Deng et al., 2016), diatoms (Passow and Alldredge, 1994; Mari and Kiorboe, 1996; Passow, 2002b), dinoflagellates (Passow and Alldredge, 1994), Prymnesiophyceae, coccolithophores included (Riebesell et al., 1995; Engel, 2004; Leblanc et al., 2009), and Cryptomonads (Kozlowski and Vernet, 1995; Passow et al., 1995). Other organisms such as *Posidonia oceanica* (Iuculano et al., 2017a), zooplankton (Passow and Alldredge, 1999; Prieto et al., 2001) and benthic suspension feeders (Heinonen et al., 2007) have also been identified as TEP producers.

TEP sources and sinks in the ocean depend not only on the taxonomic composition of TEP producers, but they are also influenced by other variables such as the organism's physiological state (Passow, 2002b), the temperature (Nicolaus et al., 1999; Claquin et al., 2008), the light (Trabelsi et al., 2008; Ortega-Retuerta et al., 2009a; Iuculano et al., 2017b), the carbon dioxide concentration (Engel, 2002), the nutrient availability (Guerrini et al., 1998; Radic et al., 2006), the turbulence conditions (Passow, 2000, 2002b) or the viral infection (Shibata et al., 1997; Vardi et al., 2012). For example, limitation by nutrients often increases TEP production, due to dissolved inorganic carbon overconsumption (Corzo et al., 2000; Engel et al., 2002a; Schartau et al., 2007), and also impedes prokaryotic consumption of TEP (Bar-Zeev and Rahav, 2015); high solar radiation can stimulate TEP production by *Prochlorococcus* during cell





decay (Iuculano et al., 2017b); and HP are known to affect TEP production by phytoplankton (Guerrini et
al., 1998; Gärdes et al., 2011) and facilitate the self–assembly of dissolved precursors into TEP (Sugimoto
et al., 2007; Ding et al., 2008).
Due to the importance of TEP in the ocean's ecology and biogeochemistry, quantifying their occurrence
across the oceans and elucidating their main distribution drivers is an essential task. It is also important to
determine the contribution of TEP as a constituent of the particulate organic carbon (POC) pool to better
understand its role in the organic matter cycling. However, in situ studies of TEP distributions in the
ocean are scarce, particularly in the open ocean (Table 2). In this study, we described the horizontal
distribution of TEP in surface waters across a North–South transect in the Atlantic Ocean, including
several biogeographical provinces in the open ocean as well as the highly productive Southwestern
Atlantic Shelf (SWAS). Our aims were (a) to identify the main biological and abiotic drivers of TEP
distribution across contrasting environmental conditions, and (b) to quantify the TEP contribution to the
entire POC pool and compare it with those of phytoplankton and heterotrophic prokaryote biomasses.

## 2 Material and methods

### 2.1 Study site and sampling

Sampling was conducted during the TransPEGASO cruise aboard the Spanish RV *Hespérides*, from 20
October to 21 November 2014. A total of 41 stations were sampled within a transit across the Atlantic
Ocean from Cartagena (SE Spain) to Punta Arenas (S Chile, Fig. 1). During the cruise, the ship crossed
five biogeographical provinces; The Northeastern Subtropical Gyre, the North Atlantic Tropical Gyre, the
Western Tropical Atlantic, the South Tropical Gyre and the SWAS (Longhurst, 1998). Seawater was
collected from 4 m depth using the ship's underway pump (BKMKC–10.11. Tecnium, Manresa, Spain).
Temperature and salinity were measured continuously using a SBE21 Sea Cat Thermosalinograph. Total
solar radiation was measured also continuously using a LI–COR Biospherical PAR Sensor. The rest of
the variables were collected twice a day (09:00:00 and 13:00:00 LT) with the ship moving at
approximately 10 knots.

### 2.2 Chemical and biological analysis

#### 2.2.1 Particulate organic matter (TEP and POC)

TEP concentrations were determined by spectrophotometry following Passow and Alldredge (1995).
Duplicate samples (100–500 mL each) were filtered through 25 mm diameter 0.4 µm pore size
Polycarbonate filters (DHI) using a constant low filtration pressure (~150 mmHg). The samples were
immediately stained with 500 µL of Alcian Blue solution (0.02 %, pH 2.5) for 5 s and rinsed with Milli–
Q water. The filters were stored frozen until further processing in the laboratory (within 8 months).
Duplicate blanks (empty filters stained as stated earlier) were prepared twice a day to correct the
interference of stained particles in TEP estimates. Both the sample and blank filters were soaked in 5 mL
of 80 % sulfuric acid for 3 h. The filters were shaken intermittently during this period. The samples were



then measured spectrophotometrically at 787 nm (Varian Cary 100 Bio). The absorbance values of filter
blanks did not change substantially between batches of samples, suggesting stability in the staining
capacity of the Alcian blue solution throughout the cruise.  The Alcian Blue solution was calibrated just
before the cruise using a standard solution of xanthan gum (XG) passed through a tissue grinder and
subsequently filtered through two sets of filters (four points in triplicate): Preweighted filters to determine
the actual concentration of the XG solution, and filters that were subsequently stained, frozen and
analysed in the spectrophotometer. The detection limit was set to 0.034 absorbance units and the mean
range between duplicates was 18.7 %. We estimated the TEP carbon content (TEP–C) using the
conversion factor of 0.51 µg TEP–C $L^{-1}$ per µg XG eq $L^{-1}$ (Engel and Passow, 2001).
POC was measured by filtering 1000 mL of seawater on pre–combusted (4 h, 450 ºC) GF/F glass fibre
filters (Whatman). The filters were stored frozen (-20 ºC) until processed. Prior to analysis, the filters
were dried at 60 ºC for 24 h in an atmosphere of HCl fumes to remove carbonates. Then filters were dried
again and analysed by high–temperature (900 ºC) combustion in an elemental analyzer (Perkin–Elmer
2400 CHN).

### 154   2.2.2 Chlorophyl *a* (Chl *a*)

Samples for fluorometric Chl *a* analyses were filtered (250 mL) on glass fibre filters (Whatman GF/F, 25
mm diameter) and stored at -20 ºC until further processing in the ship's laboratory. Pigments were
extracted with 90 % acetone at 4 ºC in the dark for 24 hours. Fluorescence of extracts was measured
according to the procedure described in Yentsch and Menzel (1963), with a calibrated Turner Designs
fluorometer. No "phaeophytin" correction was applied.

### 160   2.2.3 Inorganic nutrients

Samples for dissolved inorganic nutrients (nitrate, phosphate and silicate) were stored in 10 mL sterile
polypropylene bottles at -20 ºC until analysis. The samples were further processed in the laboratory using
standard segmented flow analyses with colorimetric detection (Hansen and Grasshoff, 1983), using a
Skalar Autoanalyzer.

### 165   2.2.4 Microscopic phytoplankton identification

We quantified phytoplankton groups by microscopy. Water was fixed with hexamine–buffered
formaldehyde solution (4 % final formalin concentration) in a glass bottle, immediately after collection,
and then was allowed to settle for 48 h in a 100 $cm^3$ composite chamber. An inverted microscope
(Utermöhl, 1958) was used to enumerate the smaller phytoplankton cells (< 20 µm, 312× magnification)
and the larger phytoplankton cells (> 20 µm, 125× magnification). Phytoplankton was identified to the
species level when possible, and finally classified into four groups: diatoms, dinoflagellates,
coccolithophores and other microplankton cells called from now on as "other microalgae". Cell C content
was calculated using conversion equations of Menden-Deuer and Lessard (2000): one for diatoms (pg C
$cell^{-1} = 0.288 \times$ volume $(\mu m^{-3})^{0.811}$) and one for the other algae groups (pg C/cell$^{-1} = 0.216 \times$ volume $(\mu m^{-3})^{0.939}$). Total carbon biomass was calculated from cell C content and cell abundance.



### 2.2.5 Picoplankton abundance

To enumerate picoplankton cells, samples (4.5 mL) were fixed with 1 % paraformaldehyde plus 0.05 % glutaraldehyde (final concentrations), let fix for 15 min. at room temperature, deep frozen in liquid nitrogen and stored frozen at -80 °C. Samples were then analysed 6 months after the cruise end, using a FACS Calibur (Becton and Dickinson) flow cytometer equipped with a 15 mW argon–ion laser emitting at 488 nm. Before analysis, samples were thawed and we added 10 µL per 600 µL sample of a $10^5$ mL$^{-1}$ solution of yellow–green 0.92 µm Polysciences latex beads as an internal standard. Samples were then run at high speed (approx. 75 µL min$^{-1}$) for 4 min. with Milli–Q water as a sheath fluid. Three groups of phytoplankton (*Prochlorococcus*, *Synechococcus* and picoeukaryotic algae) were distinguished and enumerated on the basis of the differences in their autofluorescence properties and scattering characteristics (Olson et al., 1993; Zubkov et al., 1998). Abundances were converted to biomass (µg L$^{-1}$) using average C:cell conversion factors from Simó et al. (2009): $51 \pm 18$ fg C cell$^{-1}$ for *Prochlorococcus*, $175 \pm 73$ fg C cell$^{-1}$ for *Synechococcus* and $1319 \pm 813$ fg C cell$^{-1}$ for picoeukaryotes.

### 2.2.6 Heterotrophic prokaryotic abundance (HPA)

HPA was determined by flow cytometry using the same fixing protocol and instrument as for picoplankton. Before analyses, samples were thawed, stained with SYBRGreen I (Molecular Probes) at a final concentration of 10 µM and left in the dark for about 15 min. Samples were run at a low flow rate (approximately 15 µL min$^{-1}$) for 2 min with Milli–Q water as a sheath fluid. We added 10 µL per sample of a $10^5$ mL$^{-1}$ solution of yellow–green 0.92 µm Polysciences latex beads as an internal standard. Heterotrophic prokaryotes (HP) were detected by their signature in a plot of side scatter versus FL1 (green fluorescence). HP were enumerated separately as high–nucleic–acid–containing (HNA) and low–nucleic–acid–containing cells (LNA), and the prokaryote counts presented are the sum of these 2 types. Data were gated and counted in the SSC vs FL1 plot using the BD CellQuest$^{TM}$ software. HPA was expressed in cells mL$^{-1}$. In order to estimate the numbers of HP, the numbers of cyanobacteria (*Prochlorococcus* and *Synechococcus*) measured in the same but non–stained samples were subtracted from the total number of prokaryotes counted. HPA was converted into carbon unit (prok–C) using the conversion factor of 12 fg C cell$^{-1}$ (Lee and Fuhrman, 1987; Christian and Karl, 1994).

### 2.3 Statistical analyses

We used the R software package (RStudio Team, 2016) to test for covariations and to explore the potential controlling variables of TEP distribution across the Atlantic Ocean. We performed pairwise Spearman correlation analyses between TEP and POC concentrations. We performed bivariate and multiple regression analyses (ordinary least squares, OLS) between TEP concentrations and several physical, chemical and biological variables. Data were log transformed to fulfil the requirements of parametric tests. Ranged major axis (RMA) regression would have been more suitable since there were errors in both our dependent and independent variables. However, we decided to perform OLS regressions for a better comparison of slopes between our study and those available in the literature. The non–parametric Wilcoxon–Mann–Whitney test was carried out to compare variables, like TEP and POC, among regions. Two main regions were analysed separately due to remarkable differences in nutrient, Chl




*a* and TEP concentration: the open Atlantic Ocean (OAO, n = 30), with exclusion of the single sample
from the edge of the Canary Coastal Upwelling (CU), which had a much higher TEP concentration; and
the SWAS (n = 10).

**3 Results**

**3 Results**
**3.1 TEP distribution across the surface Atlantic Ocean**
TEP concentrations ranged from 18.3 to 446.8 μg XG eq L$^{-1}$ along the entire Atlantic Ocean transect.
Across OAO, CU included, nitrate and phosphate concentrations were low and relatively homogeneous
(nitrate: 0.47 ± 0.51 μmol L$^{-1}$; phosphate: 0.11 ± 0.06 μmol L$^{-1}$). Silicate ranged between 0.20 and 1.42
μmol L$^{-1}$, and presented the minimum concentrations in the CU and surroundings, and the maximum
concentration at station 14. The temperatures ranged from 20.7 to 29.6 ºC (25.6 ± 23.8 ºC), with
maximum values in the Equatorial Counter Current (~0–20° N, 29.1–29.6 ºC), and minimum values
around the CU and in the southermost stations of the OAO (22.6–23.6 ºC). The salinity ranged between
34.8 and 37.4, with the minimum values in the Equatorial Counter Current, and the maximum values
around 10–30º S. The Chl *a* concentration was low and quite homogeneous (0.36 ± 0.22 mg m$^{-3}$), even at
the CU (0.25 mg m$^{-3}$).

In the Northeastern Subtropical Gyre (stations 1 to 7, Fig. 1) Chl *a* concentration ranged from 0.24 to 0.37
mg m$^{-3}$. The phytoplankton biomass was generally dominated by *Prochlorococcus*, with an average of
$1.68 \times 10^5 \pm 0.81 \times 10^5$ cells mL$^{-1}$, which corresponded to a biomass of 8.58 ± 4.16 μg C L$^{-1}$. TEP
concentration in this region ranged from 54.2 to 131.7 μg XG eq L$^{-1}$ (average 73.9 ± 27.3 μg XG eq L$^{-1}$).
In the station 8 we sampled the edge of the CU. The decrease in silicate (0.26 μmol L$^{-1}$) was accompanied
by a relative increase of diatoms (9.4–fold increase) and dinoflagellates (1.3–fold increase) with respect
to surrounding stations (Fig. 2, b,e). *Prochlorococcus* abundance decreased to $9 \times 10^3$ cell mL$^{-1}$ and a
biomass of 0.46 μg C L$^{-1}$. In this station, TEP concentrations were the highest found along the whole
transect (446.7 μg XG eq L$^{-1}$) but the Chl *a* concentration (0.25 mg m$^{-3}$) was lower than in the neighbour
region. Consequently the TEP:Chl *a* ratio was the highest found in the whole transect (1760.4). Moving
south, the North Tropical Gyre (stations 9 to 13) showed an increase of silicate concentration, from 0.20
to 0.79 μmol L$^{-1}$. The Chl *a* concentration ranged from 0.41 to 0.57 mg m$^{-3}$ (Fig. 2 c). In the northernmost
part of this region (stations 9 to 11), phytoplankton biomass was dominated by *Synechococcus*, with an
average of $7.7 \times 10^4 \pm 0.8 \times 10^4$ cells mL$^{-1}$, which corresponded to a biomass of 13.5 ± 1.4 μg C L$^{-1}$. By
contrast, the southernmost stations (12 and 13) were dominated by *Prochlorococcus*, with an average of
$2.6 \times 10^5 \pm 0.5 \times 10^5$ cells mL$^{-1}$, that corresponded to a biomass of 13.2 ± 2.7 μg C L$^{-1}$ (Fig. 2 e). TEP
concentrations were similar to those in the Northeastern Subtropical Gyre, ranging between 78.1 and
123.9 μg XG eq L$^{-1}$. Station 14, with a relatively high temperature (29.0 ºC) and low salinity (35.2) was
probably the most influenced by the Equatorial Counter Current. In this station, the silicate concentration
(1.41 μmol L$^{-1}$) was the maximum observed in the whole transect, and it was observed an increase of
dinoflagellates, "other microalgae" and a decrease of *Prochlorococcus*. The Chl *a* concentration (0.48



mg m$^{-3}$) was similar to the surrounding stations and TEP were 49.4 µg XG eq L$^{-1}$. Moving further south,
in the Western Tropical and the South Tropical Gyre (stations 15 to 31) Chl *a* ranged from 0.20 to 0.41
mg m$^{-3}$ and the silicate concentration decreased (0.42–1.39 µmol L$^{-1}$). TEP presented the lowest average
values of the whole transect, ranging from 25.5 to 80.4 µg XG eq L$^{-1}$. Overall in the OAO (excluding
CU), TEP ranged from 18.26 to 131.74 µg XG eq L$^{-1}$ (average 59.85 ± 27.37 µg XG eq L$^{-1}$) and the
TEP:Chl *a* ratio ranged between 81 and 360 (average 183 ± 56; Table 1).

The southernmost part of the transect corresponded to the SWAS (stations 32 to 41). In this region,
temperature (7.6–13.9 ºC) and salinity (32.6–33.6) were lower on average than those found in the OAO
(Table 1). The SWAS could be further divided into two regions according to different inorganic nutrient
(nitrate and phosphate) concentrations (p< 0.05) and phytoplankton composition. The northern SWAS
(stations 32 to 36) presented lower nitrate (0.16 to 4.15 µmol L$^{-1}$) and phosphate (0.31 to 0.62 µmol L$^{-1}$)
concentrations than the southern SWAS (stations 37 to 41; nitrate: 2.16 to 8.924 µmol L$^{-1}$, phosphate:
0.51 to 0.89 µmol L$^{-1}$). Silicate was more homogeneous throughout (0.31 to 1.27 µmol L$^{-1}$). Chl *a*
concentration across the entire SWAS (1.07–3.75 mg m$^{-3}$) was significantly higher than in the OAO, with
no major differences between the northern and the southern parts. In most of the northern SWAS,
phytoplankton biomass was dominated by "other microalgae", with an average of 10.2 × 10$^5$ ± 6.1 10$^5$
cells L$^{-1}$, which corresponded to a biomass of 43.7 ± 25.8 µg C L$^{-1}$. In station 35, an increase of diatoms
(58121 cells L$^{-1}$ and a biomass of 145.2 µg C L$^{-1}$) and dinoflagellates (44896 cells L$^{-1}$ and a biomass of
3.3 µg C L$^{-1}$) was observed, coinciding with a decrease in silicate (0.32 µmol L$^{-1}$). Here in northern
SWAS, TEP ranged from 98.6 to 427.2 µg XG eq L$^{-1}$, with the maxima in stations 34 and 35 (Fig. 2 f). In
the southern SWAS (stations 37 to 41), phytoplankton biomass was dominated by picoeukaryotes, with
an average of 6.34 × 10$^4$ ± 1.93 × 10$^4$ cells mL$^{-1}$, which corresponded to a biomass of 83.6 ± 25.5 µg C L$^{-}$
$^1$. TEP concentration ranged 168.6–395.7 µg XG eq L$^{-1}$. Overall in the SWAS, TEP ranged from 98.6 to
427.2 µg XG eq L$^{-1}$ (average 255.7 ± 130.4 µg XG eq L$^{-1}$)  and  the TEP:Chl *a* ratio ranged from 30.8 to
164.9 (average 97.2 ± 42.1) (Table 1).
**3.2 TEP contribution to the particulate organic carbon**
TEP and POC covaried significantly and positively across the entire TransPEGASO transect (Spearman
rs analysis, r = 0.91, p< 0.01, n = 17). The contribution of TEP–C to the POC pool (TEP–C%POC)
ranged between 33.5 and 103.4 % in the OAO (average 66.3 ± 18.9 %), and between 28.1 and 109.8 % in
the SWAS (average 73.2 ± 36.2 %). POC was not analysed in the CU (Fig. 3).
To better explore the importance of TEP–C with respect to other major quantifiable POC pools, we
estimated phytoplankton (phyto–C) and heterotrophic prokaryotes (prok–C) biomass throughout the
whole cruise (Fig. 2). By comparison with phyto–C and prok–C, TEP–C contributed the most to the POC
pool in both the OAO and SWAS, but not significantly in the SWAS. Phyto–C represented the second
most important POC fraction (average OAO: 32.3 %; average SWAS: 61.6 %) (Fig. 3).



### 3.3 Relationship to other variables

TEP were significantly and positively related to Chl $a$ along the entire transect ($R^2 = 0.61$, p< 0.001, n = 39, table 3). The regression equation for log converted TEP vs Chl $a$ was log TEP = 2.09 ($\pm$ 0.04) + 0.66 ($\pm$ 0.08) × log Chl $a$. Considering the two study regions separately, only in the OAO the relationship was significant, with a higher slope than in the entire transect (log TEP = 2.31 ($\pm$ 0.10) + 1.13 ($\pm$ 0.20) × log Chl $a$; $R^2 = 0.56$, p< 0.001, n = 29).

Across the whole transect, TEP presented a significant (p< 0.05) positive relationship with total phytoplankton biomass (Table 3) and with some phytoplankton biomass groups: *Synechococcus* ($R^2 = 0.30$), picoeukaryotes ($R^2 = 0.49$), diatoms ($R^2 = 0.19$) and "other microalgae" ($R^2 = 0.27$), and with HPA ($R^2 = 0.60$). TEP were negatively related to silicate ($R^2 = 0.19$) and coccolithophores ($R^2 = 0.15$).

Some differences were found considering both regions separately. Within the OAO, TEP presented a significant (p< 0.001) positive relationship with Chl $a$ ($R^2 = 0.56$), total phytoplankton biomass ($R^2 = 0.47$) and some phytoplankton groups (*Synechococcus*, picoeukaryotes, diatoms, dinoflagellates and "other microalgae", Table 3), but not with HPA. TEP showed a significant (p< 0.001) negative relationship with the previous 24 hours–averaged solar radiation ($R^2 = 0.40$). Multiple regression analyses showed the combined positive effect of Chl $a$ and HPA on TEP distribution in the OAO (Table 4). By contrast, within the SWAS, TEP only presented a significant (p< 0.05) positive relationship with total phytoplankton biomass ($R^2 = 0.62$) and HNA ($R^2 = 0.46$, Table 3).

### 4 Discussion

#### 4.1 TEP across the surface Atlantic Ocean

We present the first inventory of surface TEP concentration along a latitudinal gradient in the Atlantic Ocean, covering both open sea and shelf waters. The existing information about TEP distribution in the open sea, and particularly in the Atlantic Ocean, is restricted to areas such as the temperate northeast Atlantic Ocean (Engel, 2004; Harlay et al., 2009; Leblanc et al., 2009; Harlay et al., 2010), the northwest Atlantic Ocean (Aller et al., 2017; Jennings et al., 2017) and the tropical and subtropical Atlantic Ocean (Mazuecos, 2015; Iuculano et al., 2017b) (Table 2). TEP concentrations we measured across the OAO (CU included) study fall generally within the range reported in other studies from the open ocean (Table 2). However our levels are higher than those observed in the Mediterranean Sea (Ortega-Retuerta et al., 2010; Ortega-Retuerta et al., 2017) and the Pacific Ocean (Ramaiah et al., 2005; Kodama et al., 2014; Iuculano et al., 2017b), and lower than those reported in the Eastern Mediterranean Sea (Bar-Zeev et al., 2011).

We found maximum TEP concentrations in the regions with high nutrient supply, namely in the CU and within the SWAS. Ours are the first TEP concentrations ever measured in the SWAS (Table 1), and only three more studies have reported TEP concentrations in coastal waters of the Atlantic Ocean (Harlay et al., 2009; Harlay et al., 2010; Jennings et al., 2017). The SWAS is a high nutrient region due to the arrival





of cold rich–nutrient Subantarctic water with the Malvinas Current. This current collides near 40 ºS with
the southward flowing Brazil Current (Gordon, 1989; Piola and Gordon, 1989; Peterson and Stramma,
1991; Palma et al., 2008). The nutrient–rich water in the region is responsible for the proliferation of
phytoplankton and HP, which could explain in part the high TEP concentrations found in this region. It is
also known that large freshwater discharges are produced in the shelf (Piola, 2005), so the organic matter
input from the continent could also influence the HPA and TEP concentrations. Although no previous
information on TEP distribution exists for this area, previous studies in similarly productive areas or
during phytoplankton blooms already observed high TEP concentrations (Long and Azam, 1996; Harlay
et al., 2009; Klein et al., 2011).  The TEP levels we measured at the SWAS are generally within the range
of those reported for coastal areas (Passow and Alldredge, 1995; Passow et al., 1995; Riebesell et al.,
1995; Kiorboe et al., 1996; Hong et al., 1997; Jähmlich et al., 1998; Wild, 2000; Ramaiah et al., 2001;
Engel et al., 2002b; García et al., 2002; Radic et al., 2005; Scoullos et al., 2006; Sugimoto et al., 2007;
Harlay et al., 2009; Wurl et al., 2009; Harlay et al., 2010; Fukao et al., 2011; Klein et al., 2011; Sun et al.,
2012; Van Oostende et al., 2012; Dreshchinskii and Engel, 2017; Jennings et al., 2017). Only two studies,
in the western Baltic Sea and the Dona Paula Bay (Arabian Sea), reported TEP levels higher than ours
(Engel, 2000; Bhaskar and Bhosle, 2006).

### 4.2 TEP as an important contributor to ocean surface's particulate organic carbon

The significant positive correlation between TEP and POC observed in our study highlighted the
importance of TEP determining POC horizontal variations in the surface Atlantic Ocean, suggesting a
high contribution of TEP to this pool.  A few values of TEP–C%POC were unrealistically higher than 100
%, a feature that has also been observed in other studies (Engel and Passow, 2001; Bar-Zeev et al., 2011;
Yamada et al., 2015). This suggests the inaccuracy of the use of standard TEP–to–carbon conversion
factors (CF, 0.51 µg TEP–C $L^{-1}$ per µg Xeq. $L^{-1}$ in our case). Therefore there is a need for defining
specific CF for diverse regions or environmental conditions. Nonetheless, an alternative explanation for
the apparent oversizing of the relative TEP–C pool may be strictly methodological: TEP are determined
on filters of 0.4 µm of pore size, whereas POC is measured on glass fibre filters with nominal pore size
0.7 µm. It is plausible, thus, that part of the smaller TEP particles are not taken into account in the POC
measured.

All in all, our results clearly show that TEP–C constituted an important portion of the POC pool in the
Atlantic Ocean (from 28.1 to 109.8 %). This contribution is comparable to that reported in the Eastern
Mediterranean Sea (Bar-Zeev et al., 2011; Parinos et al., 2017), lower than in the western Arctic (Yamada
et al., 2015), but higher than in the Northeast Atlantic Ocean (Harlay et al., 2009; Harlay et al., 2010).
Both in the OAO and SWAS, TEP comprised the largest share of the POC pool, whereas phyto–C was
the second most important contributor to POC (Fig. 3). Only in one station of the SWAS phyto–C
dominated the TEP–C. The contribution of the phyto–C and prok–C to the POC pool should be taken
with caution, as the glass fibre filters (nominal pore size 0.7 µm) used to analyse POC could have not
retained all the small phytoplankton organisms and prokaryotes (Gasol and Morán, 1999), causing an
underestimation of the POC pool.



A previous study in a eutrophic system reported TEP–C as the dominant POC contributor (Yamada et al.,
2015), whereas others found that phyto–C represented the largest share to POC compared to TEP–C and
prok–C (Bhaskar and Bhosle, 2006; Ortega-Retuerta et al., 2009b; de Vicente et al., 2010). With our
results taken all together, we hypothesize that in oligotrophic conditions TEP–C is the predominant POC
fraction, because nutrient limitation favours TEP production by phytoplankton and limits TEP
consumption by bacteria. The high proportion of TEP would modify the fate of POC in the water column
(Mari et al., 2017): Since TEP are low dense particles, lower particulate matter sinking rates, or even its
accumulation in the surface layer, can be expected in this area. Conversely, in eutrophic conditions, the
predominant POC fraction depends on many variables like the community composition, the bloom stage,
and sources of TEP different from phytoplankton.

**4.3 Main drivers of TEP distribution in the surface ocean**

In order to better understand and even predict the occurrence of TEP in the surface ocean, it is important
to describe their distribution together those of their main sources (phytoplankton and heterotrophic
prokaryotes) and environmental modulators. However, most of the previous studies of TEP in the Atlantic
Ocean were restricted to local areas, and, to our knowledge, only one included a complete description of
these variables together in a long transect (Mazuecos, 2015).
Our dataset suggests that phytoplankton is the main driver of TEP distribution in the surface Atlantic
Ocean at the horizontal scale, since significant positive relationships were observed between TEP and
both Chl $a$ and phytoplankton biomass (Table 3). It is worth noting that Chl $a$ was a good estimator of
phytoplankton biomass when the entire cruise is considered, as these variables were tightly related ($R^2 =$
0.79, p–value< 0.001, n = 36). The slope of the log converted TEP–Chl $a$ relationship for the whole study
($\beta = 0.66 \pm 0.08$, Table 3) was within the upper range amongst published data (Fig. 4), and the slope in
the OAO ($\beta = 1.13 \pm 0.20$) was the highest reported so far (Table 3, Fig. 4). In the SWAS, the TEP–Chl $a$
relationship was not significant (p–value> 0.05), yet it was for TEP vs phytoplankton biomass (see
below).
The TEP:Chl $a$ ratios were significantly (p< 0.001) higher in the OAO (both including or excluding the
CU) than in the SWAS (Table 1), with the maximum value in the CU. TEP:Chl $a$ values in the OAO (CU
included) were comparable to those observed in other oligotrophic areas (Riebesell et al., 1995; García et
al., 2002; Prieto et al., 2006; Harlay et al., 2009; Ortega-Retuerta et al., 2010; Kodama et al., 2014;
Iuculano et al., 2017b; Parinos et al., 2017) (Table 2) while the values in the SWAS were comparable to
those reported in eutrophic waters (Hong et al., 1997; Ramaiah et al., 2001; Engel et al., 2002b; Corzo et
al., 2005; Ortega-Retuerta et al., 2009b). The higher TEP:Chl $a$ ratios in oligotrophic waters (Prieto et al.,
2006) are related to nutrient scarcity, because as mentioned before, it enhances TEP production by
phytoplankton and prokaryotes (Myklestad, 1977; Guerrini et al., 1998; Mari et al., 2005; Beauvais et al.,
2006). The highest TEP:Chl $a$ ratio of the entire transect observed in the CU was probably associated
with the high relative abundance of diatoms and dinoflagellates. These groups are known to be strong
TEP producers (Passow and Alldredge, 1994), and besides, previous studies have shown that TEP





production rates reach maxima at late stages of the growth cycle, once nutrients have been exhausted
(Corzo et al., 2000; Pedrotti et al., 2010; Borchard and Engel, 2015). In the CU, the relatively low Chl $a$
level along with low silicate concentrations suggests that the upwelling-triggered bloom maximum had
already passed, which resulted in a high TEP:Chl $a$ ratio. Although POC was not measured in the CU,
high TEP:Chl $a$ suggests a high proportion of TEP with respect to other particles. This would affect the
overall particle density (i.e. low density particulate matter) and the possible accumulation of TEP in the
surface of the CU, with further consequences for processes such as organic aerosol formation. In the
SWAS, the lower TEP:Chl $a$ ratios could be related with a lower rate of TEP production under relatively
replete nutrient conditions. Extending our comparison to the literature, TEP:Chl $a$ ratio is generally higher
in oligotrophic regions (Prieto et al., 2006; Ortega-Retuerta et al., 2010; Kodama et al., 2014; Iuculano et
al., 2017b) than in eutrophic regions (Hong et al., 1997; Engel et al., 2002b; Corzo et al., 2005; Ortega-
Retuerta et al., 2009b; Klein et al., 2011; Engel et al., 2017).

In the OAO, the phytoplankton groups that showed a significant ($p< 0.05$)  positive relationship to TEP
and hence were candidates to be considered as the main producers of TEP or their precursors were
*Synechococcus,* picoeukaryotes, diatoms, dinoflagellates and "other microalgae" (Table 3). All the groups
above mentioned have been reported to produce TEP (see references in the introduction). Conversely,
coccolithophores and *Prochlorococcus* did not present a significant relationship with TEP. It has been
shown in cultures that coccolithophores do not produce high amounts of TEP (Passow, 2002b), and a
previous study showed temporal disconnections between coccolithophores and TEP maxima (Ortega-
Retuerta et al., 2018). However, in a previous study in the Atlantic Ocean, Leblanc et al. (2009) found an
association of TEP with coccolithophores. The relatively high TEP production rates of *Prochlorococcus*
in culture (Iuculano et al., 2017b) were not reflected in a spatial coupling between them in our study,
suggesting different environmental modulators of TEP production by *Prochlorococcus* in the field, at
least during our cruise, since unlike our study, Mazuecos (2015) found a significant and positive
relationship of TEP concentration also with *Prochlorococcus*. In his samples, dominated by pico–
autotrophs, diatoms did not show any significant relationship.

It is remarkable that, amongst the phytoplankton groups of the present study, *Synechococcus* biomass
presented the highest relationship ($R^2 = 0.72$) with TEP concentration in the OAO. Deng et al. (2016)
demonstrated TEP production by marine *Synechococcus* in a laboratory study, but only Mazuecos (2015)
had previously found a significant and positive relationship ($R^2 = 0.26–0.36$)  between these variables in
the ocean, particularly in the Atlantic, North Pacific and Indian oceans. Mazuecos (2015) also found that
*Synechococcus* was the phytoplankton group with the highest relationship with TEP concentration. The
oligotrophic ocean covers a big portion of the global ocean and it is mostly dominated by
picophytoplankton (Agawin et al., 2000), including *Synechococcus* (Partensky et al., 1999). Our study
highlights the importance of *Synechococcus* as a TEP source in the ocean.

In the SWAS, unlike in the OAO, there was no significant relationship between any phytoplankton group
and TEP (Table 3), but there was with the total phytoplankton biomass ($R^2 = 0.62$). One of the reasons to



the lack of a relationship between TEP and any phytoplankton group seems to be the high variability of
the phytoplankton composition in the stations of SWAS, many of them capable of TEP production, which
makes difficult to discern the main group of TEP producers among phytoplankton.

Regarding the influence of abiotic factors in TEP distribution, we found a negative relationship ($R^2$ =
0.40) between TEP concentration and 24 hours–averaged solar radiation in the OAO. The OAO stations
were exposed to high solar radiation due to water transparency and their location in tropical and
subtropical regions. Ultraviolet (UV) radiation has been found to be a significant cause of TEP loss by
photolysis (Ortega-Retuerta et al., 2009a). However, it has also been proved that the solar radiation harms
picophytoplanktonic cells, inducing TEP production (Agustí and Llabrés, 2007; Iuculano et al., 2017b).
Our results suggest that the break–up of TEP by UV radiation predominates above UV stress–induced
TEP production.

The role of HP as potential drivers of TEP distribution is not straightforward, since their net effect on
TEP accumulation depends on local conditions. Across the entire transect, TEP concentration was
significantly ($p< 0.001$) and positively related to HPA (Table 3). However, the relationship was not
significant considering the regions separately, and only in the SWAS TEP were significantly ($p< 0.05$)
and positively related to HNA, considered to be a proxy of more active cells (Servais et al., 1999;
Lebaron et al., 2001). This relationship in the SWAS could indicate that HP used TEP as a significant
carbon source or that both HP and TEP were controlled by the same drivers, such as the presence of
dissolved polysaccharides, which are substrates for HP as well as TEP precursors (Mari and Kiorboe,
1996). In the OAO, despite the lack of a relationship between log converted TEP–HPA, multiple
regression analyses showed that both phytoplankton and HP contributed significantly to explain TEP
concentration variance (Table 4).

In summary, our study describes for the first time the horizontal distribution of TEP across a North–South
transect in the Atlantic Ocean. TEP constituted a large portion of the POC pool, larger than phytoplankton
at most stations and always larger than heterotrophic prokaryotic biomass. This supports the important
role of TEP in the carbon cycle. The drivers of TEP distribution were primarily phytoplankton and, to a
lesser extent, heterotrophic prokaryotes. We call for the need to carry out more extensive studies in the
ocean, both spatially and temporally, in order to better predict the occurrence of TEP and incorporate
diagnostic relationships in model projections. These diagnostic studies must be combined with further
process studies if we are to relate TEP concentrations and stickiness to important biogeochemical
processes such as microbial colonization of particles, organic matter export to the deep ocean, gas
exchange at the air–water interface and organic aerosol formation.




**Author Contribution**

M.Z. conducted the study, analysed samples, processed and analysed the data. E.O–R and R.S. designed the study and analysed data. S.N., P.R–R., M.E. and M.S. analysed samples and provided data. M.Z. wrote the manuscript with the help of all co–authors.

**Competing interests**

The authors declare that they have no conflict of interest

**Acknowledgements**

This research was funded by the Spanish Ministry of Economy and Competitiveness through projects PEGASO (CTM2012–37615) and BIOGAPS (CTM2016–81008–R) to R.S. M.Z. was supported by a FPU predoctoral fellowship (FPU13/04630) from the Spanish Ministry of Education and Culture. E.O–R was supported by a Marie Curie Actions Intra–European Fellowship (H2020–MSCA–IF–2015–703991). The authors thank Pep Gasol and Carolina Antequera for assistance with flow cytometry; Maximino Delgado for microscopic phytoplankton counts; Rocío Zamanillo and Rafael Campos for assistance with R software; and the scientists and crew on board the RV *Hespérides* for help during the cruise.





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





**Table 1.** Mean, standard deviation and range of temperature (ºC), salinity, 24 hours–averaged solar radiation (W m⁻²), nitrate (µmol L⁻¹), silicate (µmol L⁻¹), phosphate (µmol L⁻¹), Chl *a* (mg m⁻³), POC (µmol L⁻¹), HPA (× 10⁵ cells mL⁻¹), TEP (µg XG eq L⁻¹) and TEP:Chl *a* in the OAO, the edge of the Canary Coastal Upwelling (CU) and the SW Atlantic Shelf.

| | OAO | | CU | SWAS | |
|---|---|---|---|---|---|
| | Mean ± SD (ranges) | n | (n = 1) | Mean ± SD (ranges) | n |
| Temperature (ºC) | 26.0 ± 2.1 (22.6–29.6) | 30 | 23.6 | 10.7 ± 2.2  (7.6–13.9) | 9 |
| Salinity | 36.4 ± 0.6 (34.8–37.4) | 30 | 36.1 | 33.2 ± 0.3 (32.6–33.6) | 9 |
| Solar radiation 24 h (W m⁻²) | 265 ± 73  (144–362) | 26 | – | 369 ± 52  (264–425) | 10 |
| Nitrate (µmol L⁻¹) | 0.49 ± 0.53 (0.09–0.77) | 30 | 0.13 | 4.08 ± 3.08 (0.16–8.9) | 10 |
| Silicate (µmol L⁻¹) | 0.74 ± 0.27 (0.20–1.41) | 30 | 0.26 | 0.63 ± 0.35 (0.31–1.27) | 10 |
| Phosphate (µmol L⁻¹) | 0.11 ± 0.06 (0.05–0.18) | 30 | 0.16 | 0.57 ± 0.21 (0.31–0.89) | 10 |
| Chl *a* (mg m⁻³) | 0.32 ± 0.10 (0.20–0.57) | 29 | 0.25 | 2.73 ± 0.87 (1.07–3.75) | 10 |
| POC (µmol L⁻¹) | 4.2 ± 1.9 (1.7–7.1) | 12 | – | 16.6 ± 15.8 (6.8–44.3) | 5 |
| HPA (× 10⁵ cells mL⁻¹) | 7.83 ± 2.16 (4.34–14.90) | 30 | 14.56 | 29.04 ± 5.39 (13.00–70.20) | 10 |
| TEP (µg XG eq L⁻¹) | 59.8 ± 27.4 (18.3–131.7) | 30 | 446.8 | 255.7 ± 130.4 (98.6–427.2) | 10 |
| TEP:Chl *a* | 183.1 ± 55.8 (81.2–359.7) | 29 | 1760.4 | 97.2 ± 42.1 (30.8–164.9) | 10 |

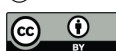

**Table 2.** Review of open–ocean surface TEP concentrations (mean and ranges; µg XG eq L[-1]), Chl $a$ (mean and ranges; mg m[-3]) and TEP:Chl $a$ ratio (mean ± SE and/or range) available in the literature. bdl: below detection limit.

| Geographic area | Sampling season | Conditions | Depth (m) | TEP mean (range) (µg XG eq. L[-1]) | Chl $a$ mean (range) (mg m[-3]) | TEP:Chl $a$ mean (range) | Reference |
|---|---|---|---|---|---|---|---|
| Fram Strait (Arctic Ocean) | Summer 2009–2012 and 2014 (time series) and summer 2014 (transect) | Bloom and non bloom | 5–150 | 75 ± 78 (5–517) | 0–4.2 | 45 ± 3–107 ± 10 | Engel et al. (2017) |
| Arctic Ocean | Autumn and Spring 2009–2010 | Sea ice covered | Above mixed layer depth | 125–1750[a] | 0.1–7.8[b] | – | Wurl et al. (2011) |
| Eastern tropical and Eastern subarctic, North Pacific Ocean | Summer 2009 | Eutrophic and oligotrophic | Above mixed layer depth | 78–970[a] | 0.3–1.7[b] | – | Wurl et al. (2011) |
| Western subarctic and North Pacific Ocean | Summer 2001 | Non bloom | 5 | 40–60 | 0.2–1.9 | – | Ramaiah et al. (2005) |
| Northeast Atlantic Ocean | Summer 1996 | Different bloom stages | 0–70 | 10[c]–124 | 0.1–1.1[c,d] | 49–104 | Engel (2004) |
| | Autumn 1996 | | 0–50 | 28.5 ± 10.2 | 0.07–0.6 | 61 | |
| Northeast Atlantic Ocean | Spring 2005 | Late stages bloom | 0–10 | 20–420[c] | 0.1–3[c,e] | – | Leblanc et al. (2009) |
| Western tropical North Pacific Ocean | Spring 2013 | Non bloom Oligotrophic | Surface mixed layer (36 ± 12) | 43 ± 7 (18–67[c]) | 0.05 ± 0.01 | 832 ± 314 | Kodama et al. (2014) |
| Western North Atlantic Ocean | Spring 2014 | Oligotrophic | 1 | 161–460 | 0.1–1[c] | – | Jennings, et al. (2017) |





| Location | Season | Trophic state | Depth | | | | Reference |
|---|---|---|---|---|---|---|---|
| Western North Atlantic Ocean and Sargasso Sea | Spring 2014 | Eutrophic and oligotrophic | 2–5 | 100–200[c] | -0.1–2.2 | – | Aller (2017) |
| Mediterranean Sea | Spring 2007 | Non bloom | Upper mixed layer | 29 (19–53) | bdl–1.8[f] | 484 (178–1293) | Ortega–Retuerta et al. (2010) |
| Western Mediterranean Sea | Spring 2012 | Oligotrophic | 0–200 | 16–25[c,g,h] | 0.1–0.7[c,h] | – | Ortega–Retuerta et al. (2017) |
| Eastern Mediterranean Sea | Winter–Autumn 2008 | Oligotrophic | 5 | 345 ±143.2 (116–420) | 0.04 ± 0.01 (0.04–0.07) | – | Bar–Zeev et al. (2011) |
| Gulf of Aqaba (Eilat, Israel) | Spring 2008 | Oligotrophic | 5 | 110–228[c] | 0.3–1.3[i] | – | Bar–Zeev et al. (2009) |
| Tropical Atlantic and Pacific Oceans | Spring–Summer 2011 | Oligotrophic | 3 | 8.18 ± 4.56 (A) 24.45 ± 2.3 (P) | 0.05–0.31 | 78.6 ± 9.3(A) 357 ± 127 (P) | Iuculano et al. (2017b) |
| Global Subtropical Atlantic, Indian and Pacific Oceans | Winter 2010– Summer 2011 | Non bloom | 0–200 | 14.0 (0.4–173.6) | 0–3[c] | – | Mazuecos (2015) |
| OAO | Autumn 2014 | Oligotrophic | 4 | 72 ± 74 (18–446) | 0.4 ± 0.2 (0.2–0.6) | 236 ± 42 (81–1760) | This study |
| OAO (CU excluded) | Autumn 2014 | Oligotrophic | 4 | 60 ± 27(18–132) | 0.3 ± 0.1 (0.2–0.8) | 183 ± 56 (81–360) | This study |
| CU | Autumn 2014 | Oligotrophic | 4 | 446 | 0.25 | 1760 | This study |
| Ross Sea | Spring 1994 | Bloom | Surface | 308 (0–2800) | 3.6 (0.3–8.8) | 85 | Hong et al. (1997) |



[a] TEP concentrations were given in $\mu$mol C L$^{-1}$. For transformation into XG units, the Engel and Passow (2001) conversion factor of 0.51 $\mu$g TEP-C L$^{-1}$ per $\mu$g XG eq L$^{-1}$ was applied.

[b] 1–8 m

[c] extracted from graphs

[d] 5 m

[e] TChl $a$

[f] 0–200 m

[g] Depth–averaged TEP

[h] stations 6–9

[i] DCM (30–40 m)

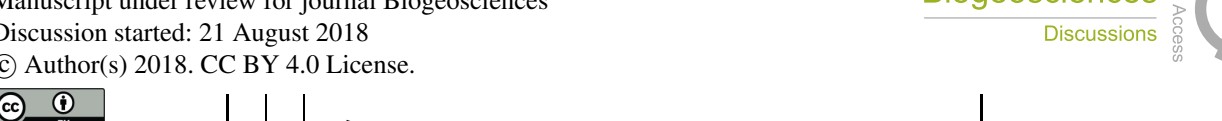

**Table 3.** Regression equations and statistics describing the relationship between TEP (dependent variable) and several independent variables throughout the TransPEGASO cruise (note all variables were $\log_{10}$–transformed). B = biomass.

| | | OAO (CU excluded) | | | | SWAS | | | | | | All | | | | |
|---|---|---|---|---|---|---|---|---|---|---|---|---|---|---|---|---|
| | | $R^2$ | $P$ | interc. | slope | n | $R^2$ | $p$ | Interc. | slope | n | $R^2$ | $p$ | Interc. | slope |
| **TEP** | SST | 0.07 | 0.16 | | | 29 | 0.06 | 0.51 | | | 9 | 0.48 | < 0.001 | 3.80 | -1.43 |
| | Salinity | 0.26 | < 0.05 | 21.78 | -12.84 | 29 | 0.002 | 0.90 | | | 9 | 0.57 | < 0.001 | 25.13 | -14.97 |
| | Solar radiation 24 h | 0.40 | < 0.001 | 4.03 | -0.96 | 26 | 0.08 | 0.44 | | | 10 | 0.03 | 0.34 | | |
| | Nitrate | 0.06 | 0.21 | | | 30 | 0.002 | 0.91 | | | 10 | 0.13 | 0.02 | 1.97 | 0.23 |
| | Phosphate | 0.04 | 0.29 | | | 30 | 0.02 | 0.69 | | | 10 | 0.37 | < 0.001 | 2.39 | 0.58 |
| | Silicate | 0.07 | 0.15 | | | 30 | 0.24 | 0.15 | | | 10 | 0.19 | < 0.005 | 1.75 | -0.80 |
| | Chl $a$ | 0.56 | < 0.001 | 2.31 | 1.13 | 29 | 0.16 | 0.24 | | | 10 | 0.61 | < 0.001 | 2.09 | 0.66 |
| | HPA | 0.04 | 0.31 | | | 29 | 0.36 | 0.06 | | | 10 | 0.60 | < 0.001 | -4.28 | 1.03 |
| | HNA | 0.01 | 0.57 | | | 29 | 0.46 | 0.03 | -0.44 | 0.46 | 10 | 0.51 | < 0.001 | -2.31 | 0.75 |
| | LNA | 0.02 | 0.43 | | | 29 | 0.02 | 0.71 | | | 10 | 0.17 | < 0.05 | -1.96 | 0.68 |
| | *Prochlorococcus* B | 0.002 | 0.80 | | | 30 | - | - | | | - | - | - | | |
| | *Synechococcus* B | 0.72 | < 0.001 | 1.72 | 0.28 | 30 | 0.005 | 0.84 | | | 10 | 0.30 | < 0.001 | 1.87 | 0.34 |
| | Ppicoeukaryotes B | 0.15 | < 0.05 | 1.68 | 0.23 | 30 | 0.005 | 0.84 | | | 10 | 0.49 | < 0.001 | 1.71 | 0.37 |
| | Diatoms B | 0.37 | < 0.001 | 2.11 | 0.28 | 27 | 0.42 | 0.058 | 2.55 | 0.16 | 9 | 0.19 | < 0.05 | 2.23 | 0.25 |
| | Dinoflagellates B | 0.18 | < 0.05 | 1.79 | 0.40 | 27 | 0.30 | 0.13 | | | 9 | 0.08 | 0.08 | | |
| | Coccolithophores B | 0.01 | 0.59 | | | 27 | 0.002 | 0.90 | | | 9 | 0.15 | < 0.05 | 1.70 | -0.23 |
| | "Other microalgae" B | 0.40 | < 0.001 | 1.75 | 0.39 | 27 | 0.0002 | 0.97 | | | 9 | 0.27 | < 0.001 | 1.86 | 0.28 |
| | Phytoplankton B | 0.47 | < 0.001 | 1.04 | 0.61 | 26 | 0.62 | < 0.05 | 0.43 | 1.00 | 9 | 0.62 | < 0.001 | 0.99 | 0.70 |

$R^2$ explained variance, n sample size, $p$ level of significance



**Table 4.** Results of multiple regression analyses between TEP (dependent variable) and combined independent variables, all $\log_{10}$–transformed. B = biomass.

| | | OAO (CU excluded) | | | | SWAS | | | | All | | | |
|---|---|---|---|---|---|---|---|---|---|---|---|---|---|
| | | Partial coefficient | Partial $p$ | $R^2$ | $p$ | Partial coefficient | Partial $p$ | $R^2$ | $p$ | Partial coefficient | Partial $p$ | $R^2$ | $p$ |
| **TEP** | Phyto B | 0.67 | < 0.001 | 0.53 | < 0.001 | 0.82 | < 0.05 | 0.66 | < 0.05 | 0.47 | < 0.01 | 0.68 | < 0.001 |
| | HPA | 0.14 | 0.58 | | | 0.38 | 0.13 | | | 0.48 | < 0.05 | | |
| | Phyto B | 0.70 | < 0.001 | 0.53 | < 0.001 | 0.76 | < 0.05 | 0.70 | < 0.05 | 0.54 | < 0.001 | 0.71 | < 0.001 |
| | HNA | 0.06 | 0.70 | | | 0.28 | 0.08 | | | 0.36 | < 0.01 | | |
| | Chl $a$ | 1.26 | < 0.001 | 0.67 | < 0.001 | 0.48 | 0.26 | 0.33 | 0.10 | 0.39 | < 0.005 | 0.66 | < 0.001 |
| | HPA | 0.56 | < 0.05 | | | 0.59 | 0.08 | | | 0.54 | < 0.01 | | |
| | Chl $a$ | 1.28 | < 0.001 | 0.60 | < 0.001 | 0.30 | 0.48 | 0.36 | 0.08 | 0.47 | < 0.001 | 0.67 | < 0.001 |
| | HNA | 0.20 | 0.20 | | | 0.42 | 0.06 | | | 0.37 | < 0.01 | | |

$R^2$ explained variance, $p$ level of significance




**Figure captions**

**Figure 1:** Hydrographic stations (filled circles) of the TransPEGASO cruise, sampled during October–November 2014 in the Atlantic Ocean. Chl *a* concentration (background color; mg m$^{-3}$) during November 2014 were taken from NASA MODIS AQUA 9–km Products composite.

5      **Figure 2:** Variations of sea surface temperature (SST, ºC) and salinity (panel (a)), nitrate, silicate and phosphate (µmol L$^{-1}$) (panel (b)), Chl *a* (mg m$^{-3}$) and POC (µmol L$^{-1}$) (panel (c)), biomass of phytoplankton and HP (µg CL$^{-1}$) (panel (d)), biomass of *Prochlorococcus*, *Synechococcus*, picoeukaryotes, diatoms, dinoflagellates, coccolithophores and "other microalgae" (µg CL$^{-1}$) (panel (e): For OAO use left axis, for SWAS use right axis) and TEP (µg XG eq L$^{-1}$) (panel (f)) in the
10    TransPEGASO cruise.

**Figure 3:** Average and standard deviation of the contribution of TEP, phytoplankton and HP to the POC pool (%) in the OAO and the SWAS.

**Figure 4:** Relationship between TEP and Chl *a* concentration from the TransPEGASO cruise, with the linear regression line (regression equation in the text). Two regions are distinguished: open Atlantic
15    Ocean (OAO, CU included, filled circles) and SW Atlantic Shelf (SWAS, empty circles). Regression lines from the literature are also shown for comparison. α and β indicate the y intercept and slope, respectively; log TEP (µg Xeq. L$^{-1}$) = α + β × log Chl *a* (mg m$^{-3}$); [a] α = 2.45 and β = 0.33, (Engel, 1998 in Passow, 2002a); [b] α = 2.25 and β = 0.65, (Hong et al., 1997); [c] α = 2.27 and β = 0.24, (Yamada et al., 2015); [d] α = 2.06 and β = 0.50, (Ramaiah and Furuya, 2002); [e] α = 1.63 and β = 0.39, (Passow and
20    Alldredge, 1995); [f] α = 1.63 and β = 0.32, (Corzo et al., 2005); [g] α = 1.08 and β = 0.38, (Ortega–Retuerta et al., 2009b).



Figure 1




Figure 2

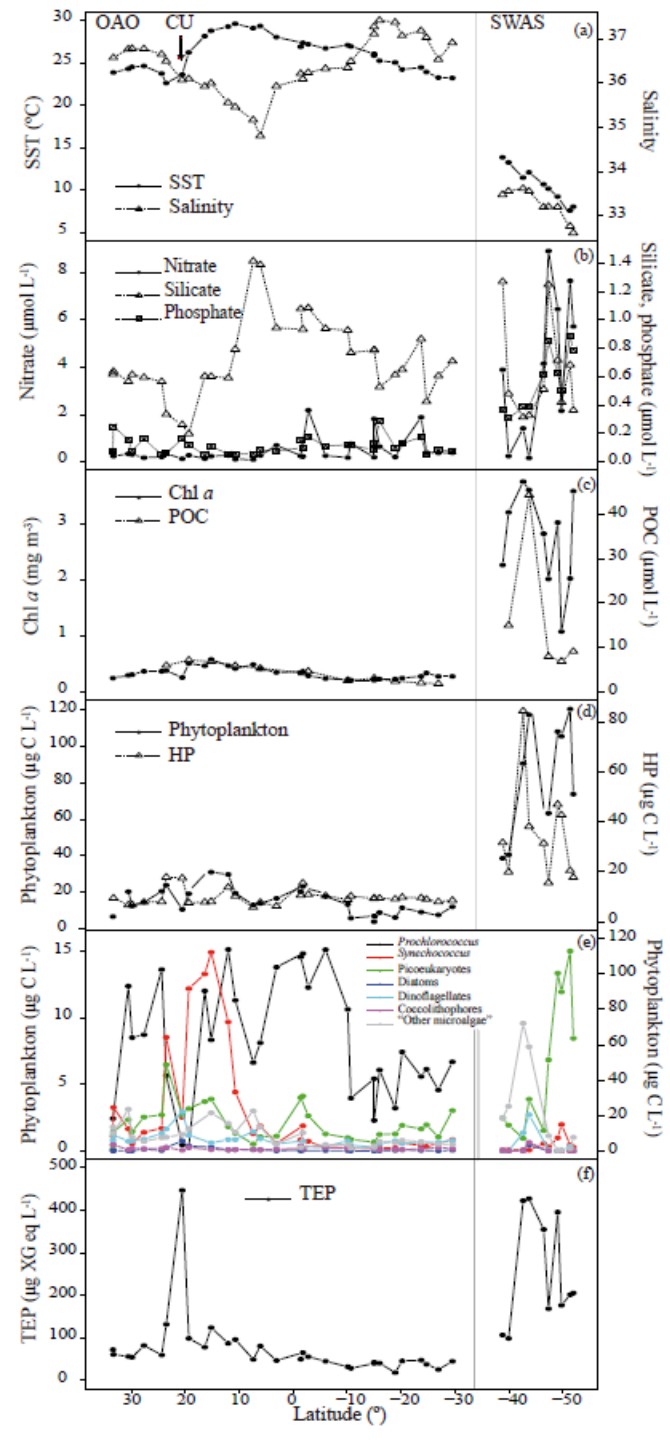



Figure 3

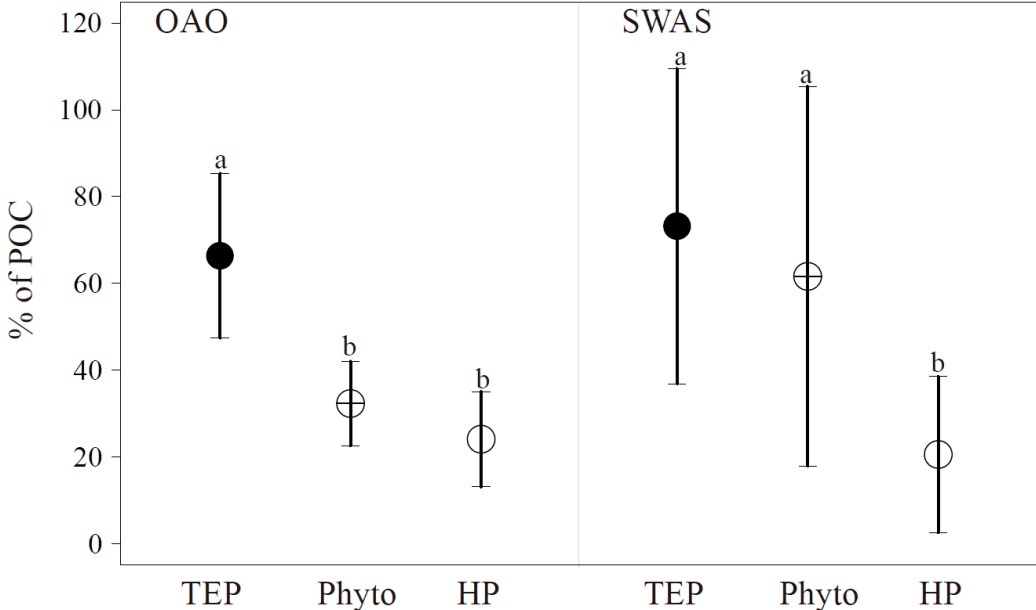




Figure 4

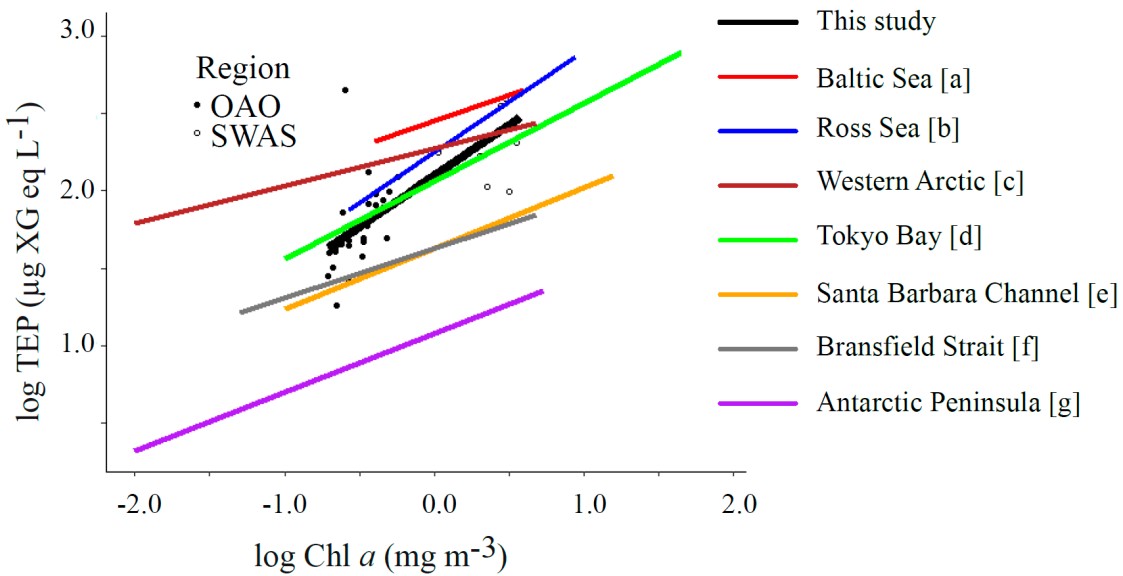