# Peer review of "the surface Atlantic Ocean"

_Biogeosciences, 2018_

## Referee Comment (RC1) · D. Kumar (Referee) · 21 Sep 2018

This is a good manuscript that provides excellent summary of TEP information. A good synthesis of data at hand despite the limitations of coverage in space (data collected only at 4m and at times the discussion is based on 1 sample to represent a hydrographic domain, say CU). The authors made the point that TEP contributes majorly to POC than phytos and HP based on the quantification of TEP, phytos and HP carbon pools estimated from available conversion factors. That the authors are well aware of limitations/approximations of these conversion factors, semi-quantitative nature estimations of TEP, phytos and HP pools (the last two are based on cell numbers) one

would have expected the authors to critically evaluate their % contributions keeping the associated overall errors (methodology+conversion). This may not alter their conclusions but convinces the readers with appropriate comparisons having taken errors into account. I recommend minor revision of this manuscript before it is accepted for publication. 1. Lines 65-66: 'Enhancing particle sinking' – The authors may want to see open ocean TEP information from North Indian Ocean (Kumar et al., 1998) 2. Line 67: 'can also ascent' gives a meaning that TEP float by themselves but these are mainly transported to surface microlayer by rising bubbles through scavenging 3. Lines 109-110: "in situ studies of TEP distributions in the ocean are scarce, particularly in the open ocean (Table 2)". But Table 2 specifies TEP in surface layers. Kumar et al. (1998) and Ramaiah et al. (2000) provided the first TEP open ocean data from the Indian Ocean (see below for references). 4. Line 115: 'entire POC' will also include non-living non-TEP organic carbon fraction. This was not addressed in the manuscript. 5. Lines 147-148: Given the 18.7% difference in concentrations between TEP duplicates specify the errors in TEP-C estimation to compare with other org C-reservoirs. 6. Lines 175 to 202 and Lines 283-287: How accurate is the cell abundances counting of the respective biological groups? Please specify uncertainties involved. This is particularly important because each of subgroups will carry uncertainties in carbon per cell and that will be additive. Total uncertainties involved assume significance since a comparison is being made with TEP-C, where TEP estimation itself is semi-quantitative! For example, line 232-233 show phytoplankton biomass estimation carries nearly 50% of uncertainties in cell counts and cell C estimations! Authors discussion (Lines 343-353) on uncertainties in TEP-C contribution to POC arising from cell-C conversion and analytical artifacts is well appreciated. But the authors should help the readers by providing a comparative evaluation including errors in estimated carbon pools in a Table. 7. Line 310: 'we present the first inventory of surface TEP concentration' – can the seawater samples collected from 4 m depth be treated as representative of surface layer to make an inventory? Here seems to be an incompatibility that needs to be clarified. 8. Lines 360-364: Given the large uncertainties involved statements such as 'Only in one

station of the SWAS phyto–C dominated the TEP–C (line 360-1)' and 'with the maximum concentrations in the edge of the Canary Coastal Upwelling (CU, n = 1) (lines 45-46)' may be avoided as these oversimplify a complex reality of spatial variability in horizontal and vertical (see line 310 comment above) dimensions. 9. Lines 367-370: A good hypothesis. 10. Line 448: Please show the negative relation in a diagram. 11. Lines 462-463: Figure 3 suggests that in spite of higher (nearly double) contribution of phytos to %POC in SWAS than in OAO, TEP and HP contributions to %POC are nearly the same. It appears that HP is more important in regulating TEP concentrations in the Atalantic, in general. This is slightly different from what has been said in lines 472-473 (The drivers of TEP distribution were primarily phytoplankton and, to a lesser extent, heterotrophic prokaryotes)

REFERENCES:

Kumar, M. D., V. V. S. S. Sarma, N. Ramaiah, M. Gauns and S. N. de Sousa (1998) Biogeochemical significance of transport exopolymer particles in the Indian Ocean. GEOPHYSICAL RESEARCH LETTERS, 25, 81-84. Ramaiah, N., V.V.S.S. Sarma, Mangesh Gauns, M. Dileep Kumar and M. Madhupratap (2000) Abundance and Relationship of Bacteria with Transparent Exopolymer Particles during the 1996 Summer Monsoon in the Arabian Sea PROCEEDINGS OF INIDAN ACADEMY OF SCIENCES (Earth Planetary Sciences), 109, 443-451.

---

## Referee Comment (RC2) · Anonymous Referee #2 · 25 Oct 2018

General comments This is a good manuscript, well written and very informative about TEP distribution in surface waters of the Atlantic Ocean, taking into account many other studies. The main limitations, in my opinion, rely on the number of data collected and the depth, only at 4 m, which probably underestimates other processes related to TEP presence especially close to the depth of the chlorophyll maximum. As pointed out by the other referee, conversions factors may be approximative and a proper critical consideration on any limitation should be included in the manuscript. Another general recommendation: TEP importance in processes such as air-sea gas exchange, aerosol formation, marine snow and carbon export and cycling should be better addressed in the whole study, as focal points of TEP influence on carbon dynamics, please see my

comments in the introduction and discussion. I recommend that the following points are addressed before publication.

Abstract: Lines 37-38: The authors should be aware that air-sea gas exchange and aerosol emissions are complex processes, which are not properly explained in the manuscript. I would thus remove this sentence that in the abstract appears a bit vague and would concentrate on the role of TEP in channeling the carbon produced by primary productivity (see Mari et al. 2017). Lines 50-51: it could also be an inhibited TEP-aggregation by UV, not just breaking. I would rephrase the sentence.

Introduction: Lines 63-75: The introduction is a bit vague, I would introduce the concept of a marine gel, the composition and cross-links in the molecule that make TEP water insoluble but still subject to fragmentation and further aggregation processes, and the size distribution in the ocean, mentioning the size range we are talking about. 0.4 $\mu$m falls into the truly dissolved phase, and a discussion on the continuum of sizes linking DOM and POM should be added. Several species can directly release TEP or macrogels, but such macromolecules can also form from dissolved abiotic material in the absence of phytoplankton (Chin, W.-C., Orellana, M.V., Verdugo, P., 1998. Spontaneous assembly of marine dissolved organic matter into polymer gels. Nature 391, 568–572.) Moreover, the importance of TEP and marine snow should be mentioned. The role of TEP in the sea-surface microlayer should be either expanded or left out. The description presented here about air-sea gas exchange and aerosol is a bit vague and not precise. I would suggest spending more words on it, especially because 4 m depths is close to the surface so surface ocean processes and air-sea interaction should be properly mentioned. The role of TEP in the sea-surface microlayer depends on many factors: wind speed, primary productivity, and they are not the only class of gel particles present (e.g., highly productive region, see Engel and Galgani 2016, Biogeosciences, Wurl, O., Miller, L., Röttgers, R., and Vagle, S.: The distribution and fate of surface-active substances in the sea-surface microlayer and water column, Mar. Chem., 115, 1–9, 2009. Wurl, O., Miller, L., and Vagle, S.: Production and fate

of trans- parent exopolymer particles in the ocean, J. Geophys. Res., 116, C00H13, doi:10.1029/2011JC007342, 2011. ). Line 69: specify what do you mean by "affect air-sea gas exchange". Lines 70-71: caution is needed here. Orellana et al. discuss about micro and nanogels, determined with a different method with respect to the one reported here. When gels are present in the sea-surface microlayer, it will depend on their size distribution whether they will be part of the organic aerosol fraction or not. Aerosol particles smaller than 1 $\mu$m will be part of the accumulation mode of sea-spray aerosols, but when further aggregating and reaching sizes above 2.5 $\mu$m they won't actually stay in the atmosphere longer than a few hours – as their size distribution is described as coarse mode aerosols. TEP as macromolecules are between accumu- lation and coarse mode but not ice-nucleating particles or cloud condensation nuclei. Another consideration is that if high wind speed are present (above 5 m/s), there might be increased aggregation rates of TEP with solid particles which will favour the forma- tion of negatively buoyant aggregates that will sink out of the surface microlayer and surface waters in general. Line 80: not just photolysis but also UV inhibited aggregation of precursor polymers limits TEP formation. Line 102: What does this sentence mean? Please explain how HP affect TEP production and assembly of precursors. Line 106: I suggest introducing the concept of biological carbon pump and the importance of TEP in ocean carbon cycle, as this is a central idea of the study. How much estimated primary production carbon is channeled into the TEP pool? (See Mari et al., 2017). This could also help making confrontations with phytoplankton-derived carbon, still es- timates but could be interesting. Lines 107-108: As mentioned already, TEP span over a wide range - DOC or POC is just an operational definition. From colloids (dissolved) to macrogels (particulate) (see Verdugo 2012 Annual Rev. of marine sciences).

Methods: If you have DOC data, I think it would be worth showing them and looking for the missing fraction that drives POC underestimation with respect to TEP, as TEP are connecting both pools of organic matter. Can you provide a standard deviation or error estimation for POC filters? Do you have wind speed information? This would be useful in estimating whether TEP could accumulate in the surface layer.

Discussion: Lines 317-319: Can you provide any reason why you think your values are higher than those observed in the Mediterranean Sea and Pacific Ocean? Is it related to nutrient concentration/time of year, different analysis method (e.g. spectroscopy vs microscopy for gel particles identification), depth?

Lines 355-356: The authors should mention here any limitation of the conversion factors. Line 331: is the organic matter that influences HPA concentration and their TEP production or ..? How does the organic matter pool influences TEP formation? If you mean, by abiotic assembly of a pool of dissolved precursors, this concept should be mentioned early in the introduction.

Lines 370-374: Again, the fate of TEP depends on further aggregation processes. Generally less dense than water could accumulate in the surface microlayer but wind speeds, high heterotrophic activity, coagulation with other organic and mineral particles thanks to their stickiness should be mentioned to describe their fate in the area. Which one do you think would predominate?

Lines 409-410: Again on aerosol formation, it's a complex process and without any information on the size distribution of TEP in this study I would recommend caution in making such affirmations. Here, it's a bit a "stand alone" sentence without any further explanation, which does not make much sense. It should be expanded and explained better. Also, please see the paper by Quinn et al. 2014 "Contribution of sea surface carbon pool to organic matter enrichment in sea spray aerosol" Nature Geoscience, which actually breaks up the concept of organic aerosols related to phytoplankton blooms (and in this case, TEP). Line 444: as mentioned, TEP can also be produced by aggregation of colloids in the absence of phytoplankton, that is, in the presence of polymeric precursors in the dissolved phase. Thus, it would be interesting to see the relationship of TEP to DOC or acidic sugars.

Line 454: UV also inhibits gel aggregation (Orellana and Verdugo 2003, Ultraviolet radiation blocks the organic carbon exchange between the dissolved phase and the

gel phase in the ocean, Limnology and Oceanography). It should be mentioned here.

---

## Author Comment (AC1) · 23 Nov 2018

We thank the reviewer for his valuable comments on our manuscript. We answer below to each comment and question. (a more edited version is submitted in pdf)

R: This is a good manuscript that provides excellent summary of TEP information. A good synthesis of data at hand despite the limitations of coverage in space (data collected only at 4m and at times the discussion is based on 1 sample to represent a hydrographic domain, say CU).

A: We thank the reviewer for his supportive comments. We were not trying to repre-

sent a hydrographic domain with a single sample, but we just treated the CU as an independent sample (i.e. removing it when calculating TEP averages and regression analyses between TEP and other environmental and biological parameters) due to its particularity, as indicated in the objectives section (end of the introduction). We are aware that some sentences could have given that impression and we will fix this in the revised version of the MS. For example we will change the following sentences:

Line 45: "with the maximum concentrations in the SWAS and in a station located at the edge of the Canary Coastal Upwelling (CU)"

Line 223: "and presented the minimum concentrations in the CU station and surroundings"

Line 322: "namely in the station located in the CU and within the SWAS"

Line 392: "with the maximum value in the station located in the CU"

Line 401: "The highest TEP:Chl a ratio of the entire transect observed in the station located in the CU was probably associated with the high relative abundance of diatoms and dinoflagellates."

R: The authors made the point that TEP contributes majorly to POC than phytos and HP based on the quantification of TEP, phytos and HP carbon pools estimated from available conversion factors. That the authors are well aware of limitations/approximations of these conversion factors, semi-quantitative nature estimations of TEP, phytos and HP pools (the last two are based on cell numbers) one would have expected the authors to critically evaluate their % contributions keeping the associated overall errors (methodology+conversion). This may not alter their conclusions but convinces the readers with appropriate comparisons having taken errors into account. I recommend minor revision of this manuscript before it is accepted for publication.

A: We will add information in the manuscript regarding the errors associated to the methodology and conversion factors. More specifics are given in the responses below.

R: 1. Lines 65-66: 'Enhancing particle sinking' – The authors may want to see open ocean TEP information from North Indian Ocean (Kumar et al., 1998)

A: We will add the suggested reference in the revised version of the MS.

R: 2. Line 67: 'can also ascent' gives a meaning that TEP float by themselves but these are mainly transported to surface microlayer by rising bubbles through scavenging

A: We will change the sentence to: "On their way to aggregation, and due to their low density, TEP and TEP–rich microaggregates formed near the surface may ascend and accumulate in the sea surface microlayer (SML) (Engel and Galgani, 2016), a process that is largely enhanced by bubble-associated scavenging (Azetsu-Scott and Passow, 2004; Wurl et al., 2009; Wurl et al., 2011b)."

R: 3. Lines 109-110: "in situ studies of TEP distributions in the ocean are scarce, particularly in the open ocean (Table 2)". But Table 2 specifies TEP in surface layers. Kumar et al. (1998) and Ramaiah et al. (2000) provided the first TEP open ocean data from the Indian Ocean (see below for references).

A: We thank the reviewer for drawing our attention to these references. We will specify in the text and figure legend that we are referring to surface measurements. Note that, for the sake of direct comparison with our study, Table 2 only listed TEP measurements conducted with the spectrophotometric method and Xanthan Gum calibration. However, in order to be more inclusive, we will add the indicated references.

We will add the following information to Table 2: [see Figure 1 attached]

R: 4. Line 115: 'entire POC' will also include non-living non-TEP organic carbon fraction. This was not addressed in the manuscript.

A: We are aware that POC also includes other organic particle fractions such as non-living non-TEP organic carbon (for instance, cell fragments and proteinaceous - Coomassie stainable particles). In the present work we decided to compare our target variable, TEP, with the two pools of POC that are considered most abundant in sea

water, namely phytoplankton and heterotrophic prokaryotes. We will add the following sentence in the results to clarify it: "To better explore the importance of TEP–C with respect to other major quantifiable POC pools, we estimated phytoplankton biomass (phyto–C) and HP biomass (HP–C) throughout the whole cruise (Fig. 2). It is worth mentioning that POC also includes other fractions of non-living non-TEP organic carbon (e.g., cell fragments and Coomassie stainable particles), but phytoplankton and heterotrophic prokaryotes are generally considered the most abundant in open sea water (Ortega-Retuerta et al., 2009b; Yamada et al., 2015). TEP–C contributed the most to the POC pool in the OAO, where it represented twice the share of phyto-C and HP-C. In the SWAS, conversely, TEP-C was not significantly different than phyto-C, and three times higher than HP-C (Fig. 3)."

R: 5. Lines 147-148: Given the 18.7% difference in concentrations between TEP duplicates specify the errors in TEP-C estimation to compare with other org C-reservoirs.

A: We will specify the errors of TEP-C estimations in the material and methods of the revised version of the MS:

Line 146: "We estimated the TEP carbon content (TEP–C) using the conversion factor of 0.51 $\mu$g TEP–C L-1 per $\mu$g XG eq L-1 (Engel and Passow, 2001). Errors in TEP-C estimations averaged 8.4 $\mu$g C L-1 (0.2- 70.3 $\mu$g C L-1)."

R: 6. Lines 175 to 202 and Lines 283-287: How accurate is the cell abundances counting of the respective biological groups? Please specify uncertainties involved. This is particularly important because each of subgroups will carry uncertainties in carbon per cell and that will be additive. Total uncertainties involved assume significance since a comparison is being made with TEP-C, where TEP estimation itself is semi-quantitative! For example, line 232-233 show phytoplankton biomass estimation carries nearly 50% of uncertainties in cell counts and cell C estimations! Authors discussion (Lines 343-353) on uncertainties in TEP-C contribution to POC arising from cell-C conversion and analytical artifacts is well appreciated. But the authors should

help the readers by providing a comparative evaluation including errors in estimated carbon pools in a Table.

A: Replicates for the prokaryotic abundance measurement with flow cytometry were not done because the standard errors obtained are usually very low (i.e around 1.5 % in Pernice et al. 2015).

Line 199: "Only one replicate was analysed since standard errors of duplicates are usually very low (around 1.5 % in Pernice et al., 2015)."

Microscopic observations must be interpreted with caution due to the following (Kozlowski et al., 2011; Cassar et al., 2015): - They are biased towards relatively large forms (> 5 $\mu$m) of phytoplankton groups with identifiable morphological characteristics - Problems associated with biovolume estimates - Problems with the microscopic identification of naked and small-celled groups

Line 175: "Uncertainty sources for micro-phytoplankton biomass estimates are the conversion factors, biovolume estimates, and proper identification based on morphological characteristics, harder for naked cells and those at the lower size edge (5-10 $\mu$m) (Kozlowski et al., 2011; Cassar et al., 2015)."

Regarding the phytoplankton biovolume-to-carbon conversion factors, we show the 95 % confidence intervals obtained by Menden-Deuer and Lessard (2000) for phytoplankton biomass estimation: log pg C cell-1=log a (95 % C.I.) + b (95 % C.I.) x log V ($\mu$m3), where log a is the y-intercept, b is the slope and 95% C.I. is the 95 % confidence intervals:

Protist plankton: log pg C cell-1=log -0.665 (0.132) + 0.939 (0.041) x log V ($\mu$m3)
Diatoms: log pg C cell-1=log -0.541 (0.099) + 0.811 (0.028) x log V ($\mu$m3)

Line 173: "Cell C content was calculated using conversion equations of Menden-Deuer and Lessard (2000) log pg C cell-1=log a (95 % confidence intervals) + b (95 % confidence intervals) x log V ($\mu$m3): one for diatoms (log pg C cell-1=log -0.541 (0.099)

[Figure]

+ 0.811 (0.028) x log volume ($\mu$m3)) and one for the other algae groups (log pg C cell-1=log -0.665 (0.132) + 0.939 (0.041) x log volume ($\mu$m3))."

As for the bacterial cell-to-carbon conversion factor, we will add the following explanation: Line 202: "Ducklow (2000) summarized the carbon contents of free-living marine bacteria reported in the literature for a number of oceanic regions, bays and estuaries. The average $\pm$ standard deviation for open ocean regions was 12.3 $\pm$ 2.5 fg C cell-1. A factor of 12 fg C cell-1 is equivalent to use the empirical equation proposed by Norland (1993), fgC cell-1 = 0.12 ($\mu$m3 cell vol)0.72, for an average bacterial biovolume of 0.04 $\mu$m3."

In relation to line 232-233: "The phytoplankton biomass was generally dominated by Prochlorococcus, with an average of 233 1.68 $\times$ 105 $\pm$ 0.81 $\times$ 105 cells mL-1, which corresponded to a biomass of 8.58 $\pm$ 4.16 $\mu$g C L-1.", the standard deviation of biomass is not the uncertainty of the estimate, but the variability (standard deviation) of biomass along the Northeastern Subtropical Gyre.

R: 7. Line 310: 'we present the first inventory of surface TEP concentration' – can the seawater samples collected from 4 m depth be treated as representative of surface layer to make an inventory? Here seems to be an incompatibility that needs to be clarified.

A: We agree with the reviewer that 4 m may at times not be representative of surface waters. Relatively high variability within the top surface meters has sometimes been observed (Wurl et al., 2009). However, 4 meters is usually considered as surface in most oceanographic studies, where sampling is mostly conducted either with the CTD rosette or with an underway pumping system. Nonetheless, the word 'inventory' may induce misunderstanding, and we will change it to 'distribution'. We will modify line 310-311 of the manuscript as follows:

"We present the first distribution of surface (4 m) TEP concentration along a latitudinal gradient in the Atlantic Ocean, covering both open sea and shelf waters. It is worth

mentioning that vertical variability within the top surface meters (< 4 m) has sometimes been observed (Wurl et al., 2009), but 4 m is usually considered "surface ocean" in studies where samples are collected with either an oceanographic rosette or an under-way pumping system."

R: 8. Lines 360-364: Given the large uncertainties involved statements such as 'Only in one station of the SWAS phyto–C dominated the TEP–C (line 360-1)' and 'with the maximum concentrations in the edge of the Canary Coastal Upwelling (CU, n = 1) (lines 45-46)' may be avoided as these oversimplify a complex reality of spatial variability in horizontal and vertical (see line 310 comment above) dimensions.

A: As explained above, we were not trying to represent a hydrographic domain with a single sample, so we will make the appropriate changes in the text to clarify that we are just referring to our dataset without any purpose to generalise. E.g.: Line 45-6: "with the maximum concentrations in the station located in the edge of the Canary Coastal Upwelling (CU) and the SWAS"

R: 10. Line 448: Please show the negative relation in a diagram.

A: We will add this plot to the supplementary section. [see figure 2 attached]

Figure S1. Relationship between the accumulated (previous 24 hours-average) solar irradiance (W m-2) and TEP ($\mu$g XGeq. L-1) in the OAO. The linear regression line is plotted and the equation indicated.

R: 11. Lines 462-463: Figure 3 suggests that in spite of higher (nearly double) contribution of phytos to %POC in SWAS than in OAO, TEP and HP contributions to %POC are nearly the same. It appears that HP is more important in regulating TEP concentrations in the Atlantic, in general. This is slightly different from what has been said in lines 472-473 (The drivers of TEP distribution were primarily phytoplankton and, to a lesser extent, heterotrophic prokaryotes)

A: The identification of drivers of TEP distribution is based on regression analyses of

covariation (Table 3). In OAO, the largest share of TEP variance is explained by Chl a (R2=0.56) and phytoplankton biomass (0.47), particularly Synechococcus biomass (0.72), and in the SWAS it is phytoplankton biomass (0.62) followed by High nucleic acid containing prokaryotic heterotrophs (0.46). The fact that phytoplankton mainly drive TEP variability despite very different contribution to total POC is further exemplified by the large difference in the TEP:Chl a ratio between the two regions. In other words, the two regions are characterized by phytoplankton differently prone to TEP production, but in both phytoplankton are the main TEP drivers.

REFERENCES

Cassar, N., Wright, S. W., Thomson, P. G., Trull, T. W., Westwood, K. J., de Salas, M., Davidson, A., Pearce, I., Davies, D. M., and Matear, R. J.: The relation of mixed-layer net community production to phytoplankton community composition in the Southern Ocean, Global Biogeochemical Cycles, 29, 446-462, 10.1002/2014gb004936, 2015.

Ducklow, H.: Bacterial production and biomass in the oceans, in: Microbial Ecology of the Oceans, edited by: Kirchman, D. L., 2000.

Kozlowski, W. A., Deutschman, D., Garibotti, I., Trees, C., and Vernet, M.: An evaluation of the application of CHEMTAX to Antarctic coastal pigment data, Deep Sea Research Part I: Oceanographic Research Papers, 58, 350-364, 10.1016/j.dsr.2011.01.008, 2011.

Menden-Deuer, S., and Lessard, E. J.: Carbon to volume relationships for dinoflagellates, diatoms, and other protist plankton, Limnology and Oceanography, 45, 569-579, 2000.

Norland, S.: The relationship between biomass and volume of bacteria, in: Handbook of methods in aquatic microbial ecology, edited by: Publishers, L., 303-307, 1993.

Ortega-Retuerta, E., Reche, I., Pulido-Villena, E., Agustí, S., and Duarte, C. M.: Uncoupled distributions of transparent exopolymer particles (TEP) and dissolved carbohydrates in the Southern Ocean, Marine Chemistry, 115, 59-65, 10.1016/j.marchem.2009.06.004, 2009.

Pernice, M. C., Forn, I., Gomes, A., Lara, E., Alonso-Saez, L., Arrieta, J. M., del Carmen Garcia, F., Hernando-Morales, V., MacKenzie, R., Mestre, M., Sintes, E., Teira, E., Valencia, J., Varela, M. M., Vaque, D., Duarte, C. M., Gasol, J. M., and Massana, R.: Global abundance of planktonic heterotrophic protists in the deep ocean, ISME J, 9, 782-792, 10.1038/ismej.2014.168, 2015.

Wurl, O., Miller, L., Röttgers, R., and Vagle, S.: The distribution and fate of surface-active substances in the sea-surface microlayer and water column, Marine Chemistry, 115, 1-9, 10.1016/j.marchem.2009.04.007, 2009.

Yamada, Y., Fukuda, H., Uchimiya, M., Motegi, C., Nishino, S., Kikuchi, T., and Nagata, T.: Localized accumulation and a shelf-basin gradient of particles in the Chukchi Sea and Canada Basin, western Arctic, Journal of Geophysical Research: Oceans, 120, 4638-4653, 10.1002/2015jc010794, 2015.

Please also note the supplement to this comment:
https://www.biogeosciences-discuss.net/bg-2018-359/bg-2018-359-AC1-supplement.pdf

| Sargasso Sea | Oligotrophic | Spring, summer, autumn 2012 and spring 2013 | 0–100 | 21 ± 2– 57 ± 3 | 0.05–1 [c] | | Cisternas–Novoa et al. (2015) |
|---|---|---|---|---|---|---|---|
| North Indian Ocean -Arabian Sea -Bay of Bengal | -Eutrophic | -August 1996 -September 1996 | 0–1000 | -60 [j,k] (<5–102 [j]) -7–13 [c,j] | | | Kumar et al., (1998), Ramaiah et al., (2000) |

j: TEP concentrations were given in milligram equivalent of alginic acid L$^{-1}$ and absorbance was measured at 745 nm instead of 787 nm

k: 0–50 m

**Fig. 1.**

[Figure]

**Fig. 2.**

---

## Author Comment (AC2) · 23 Nov 2018

(a more edited version is submitted as pdf)

General comments

R: This is a good manuscript, well written and very informative about TEP distribution in surface waters of the Atlantic Ocean, taking into account many other studies. The main limitations, in my opinion, rely on the number of data collected and the depth, only at 4 m, which probably underestimates other processes related to TEP presence especially close to the depth of the chlorophyll maximum.

[Figure]

A: We thank the reviewer for his/her positive general comments. We fully agree that it would be more informative to show vertical TEP profiles within the euphotic layer, but we consider that the study, which was carried out during a transit (i.e. no opportunities for CTD stations), adds valuable information for many processes happening at the ocean surface, as pointed out in the introduction.

R: As pointed out by the other referee, conversions factors may be approximative and a proper critical consideration on any limitation should be included in the manuscript.

A: We will add information about the uncertainty of phytoplankton and heterotrophic prokaryotes biomass estimate (see Referee#1).

R: Another general recommendation: TEP importance in processes such as air-sea gas exchange, aerosol formation, marine snow and carbon export and cycling should be better addressed in the whole study, as focal points of TEP influence on carbon dynamics, please see my comments in the introduction and discussion. I recommend that the following points are addressed before publication.

Abstract:

R: Lines 37-38: The authors should be aware that air-sea gas exchange and aerosol emissions are complex processes, which are not properly explained in the manuscript. I would thus remove this sentence that in the abstract appears a bit vague and would concentrate on the role of TEP in channeling the carbon produced by primary productivity (see Mari et al. 2017).

A: We appreciate the reviewer's comment. We will remove the sentence about aerosol emission and change it. We will mention the role of TEP channeling the biological pump.

Line 36: "Transparent exopolymer particles (TEP) are a class of gel particles, produced mainly by microorganisms, which play important roles in biogeochemical processes such as carbon cycling and export. TEP (a) are colonized by carbon-consuming microbes; (b) mediate aggregation and sinking of organic matter and organisms, thereby contributing to the biological carbon pump; and (c) accumulate in the surface microlayer (SML) and affect air-sea gas exchange."

Lines 50-51: it could also be an inhibited TEP-aggregation by UV, not just breaking. I would rephrase the sentence.

We will change this sentence in the revised version of the MS as follows: "suggesting that sunlight, particularly UV radiation, is more a sink than a source for TEP. "

We will also mention it in the introduction (see comments below), and the discussion:

Line 451: "Ultraviolet (UV) radiation causes TEP loss by photolysis (Ortega-Retuerta et al., 2009a) and inhibits TEP formation from precursors (Orellana and Verdugo, 2003."

Line 454: "Our results suggest that the roles of UV radiation in breaking up TEP and/or limiting their formation from precursors overcome UV stress–induced TEP production."

Introduction:

A: We thank the reviewer for his/her thorough effort to improve the introduction section of our manuscript.

Lines 63-75: The introduction is a bit vague, I would introduce the concept of a marine gel, the composition and cross-links in the molecule that make TEP water insoluble but still subject to fragmentation and further aggregation processes, and the size distribution in the ocean, mentioning the size range we are talking about. 0.4 $\mu$m falls into the truly dissolved phase, and a discussion on the continuum of sizes linking DOM and POM should be added.

We will add the concept of DOM-POM continuum and evoke the gel polymer theory introducing a sentence like this in the revised version of the MS: "TEP are gel-like substances mainly formed by the spontaneous assembly from dissolved precursors,

namely some acidic polysaccharides, which are stabilized as TEP either by covalent links or ionic strength. Therefore, the formation and fragmentation of TEP from/to dissolved precursor material spans the dissolved to particulate continuum of organic matter in the sea". However, one could consider 0.4 $\mu$m as a fraction included in the particulate phase if the 0.2 $\mu$m cutoff (one most widely used) is taken into account.

Line 80: "Regarding the sources, TEP are released by organisms, mainly microorganisms, during production and decomposition processes, either directly as detritus (Hong et al., 1997; Berman-Frank et al., 2007), or indirectly as dissolved precursors that can self-assemble to form TEP (operationally defined as particles > 0.4 $\mu$m) (Passow and Alldredge, 1994; Chin et al., 1998; Thuy et al., 2015) TEP are stabilized by covalent links or ionic strength (Cisternas-Novoa et al., 2015) and therefore, the formation and fragmentation of TEP from/to dissolved precursor material spans the dissolved to particulate continuum of organic matter in the sea."

R: Several species can directly release TEP or macrogels, but such macromolecules can also form from dissolved abiotic material in the absence of phytoplankton (Chin, W.-C., Orellana, M.V., Verdugo, P., 1998. Spontaneous assembly of marine dissolved organic matter into polymer gels. Nature 391, 568–572.)

A: We already mentioned this process in line 82 but we will rephrase it in order to clarify the spontaneous assembly of DOM into TEP (See comment above).

R: Moreover, the importance of TEP and marine snow should be mentioned. The role of TEP in the sea-surface microlayer should be either expanded or left out. The description presented here about air-sea gas exchange and aerosol is a bit vague and not precise. I would suggest spending more words on it, especially because 4 m depths is close to the surface so surface ocean processes and air-sea interaction should be properly mentioned. The role of TEP in the sea-surface microlayer depends on many factors: wind speed, primary productivity, and they are not the only class of gel particles present (e.g., highly productive region, see Engel and Galgani

2016, Biogeosciences, Wurl, O., Miller, L., Röttgers, R., and Vagle, S.: The distribution and fate of surface-active substances in the seasurface microlayer and water column, Mar. Chem., 115, 1–9, 2009. Wurl, O., Miller, L., and Vagle, S.: Production and fate of transparent exopolymer particles in the ocean, J. Geophys. Res., 116, C00H13, doi:10.1029/2011JC007342, 2011).

A: We will add a few sentences to better introduce why TEP accumulates in the sea surface, implications and factors affecting this accumulation. Lines 63-75: "Transparent exopolymer particles (TEP) are defined as a class of non–living organic particles in aqueous media, mainly formed by acidic polysaccharides, that are stainable with Alcian Blue (Alldredge et al., 1993). Due to their stickiness, TEP favour the formation of large aggregates of organic matter and organisms (typically named marine snow), enhancing particle ballast and sinking in the ocean (Logan et al., 1995; Kumar et al., 1998; Passow et al., 2001; Burd and Jackson, 2009). The presence of TEP also affects the microbial food–web, as they can be used as a food source for zooplankton (Decho and Moriarty, 1990; Dilling et al., 1998; Ling and Alldredge, 2003) and heterotrophic prokaryotes (HP) (Passow, 2002b) through microbial colonization of aggregates (Alldredge et al., 1986; Grossart et al., 2006; Azam and Malfatti, 2007). On their way to aggregation, and due to their low density, TEP and TEP–rich microaggregates formed near the surface may ascend and accumulate in the sea surface microlayer (SML) (Engel and Galgani, 2016), a process that is largely enhanced by bubble-associated scavenging (Azetsu-Scott and Passow, 2004; Wurl et al., 2009; Wurl et al., 2011). This accumulation in the SML, also contributed by local direct production (Wurl et al., 2011) can supress the air-sea exchange of $CO_2$ and other trace gases by acting as a physicochemical barrier or modifying sea surface hydrodynamics at low wind speeds (Calleja et al., 2008; Cunliffe et al., 2013; Wurl et al., 2016). Sea surface TEP can also be released to the atmosphere by bubble bursting (Zhou et al., 1998; Aller et al., 2005; Kuznetsova et al., 2005), contributing to organic aerosol and possibly acting as cloud condensation nuclei and ice nucleating particles (Orellana et al., 2011; Leck et al., 2013; Wilson et al., 2015). All in all, TEP play important roles in microbial diversity, carbon cycling, and

carbon exports to both the deep ocean and the atmosphere."

R: Line 69: specify what do you mean by "affect air-sea gas exchange".

A: Some studies, revised in Cunliffe et al. (2013), show the influence of surface active components of the SML (including biogenic polysaccharides) on air-sea gas exchange, either acting as a physicochemical barrier or modifying sea surface hydrodynamics, which in turn results in a suppression of air-water gas exchange. For example, Calleja et al. (2008) found that the organic matter content of the surface water supressed $CO_2$ gas exchange between the air and the ocean at low and intermediate wind speeds (> 5 m s-1). Wurl et al. (2016) found enrichments of TEP, POC, PON, total prokaryotic cell numbers and picophytoplankton abundances in sea microlayers at multiple stations of different regions, compared to the underlying bulk water, being higher in slick surfaces than non-slick ones, and estimated that slicks could reduce $CO_2$ fluxes by up to 15 %, which highlight the importance of slicks in regulating air-sea interactions. Jenkinson et al. (2018) reviewed recently known and suspected mechanical aspects of how biologically produced organic matter modulates air-sea fluxes of $CO_2$.

We will briefly add some of this information in the introduction section. See comment above.

R: Lines 70-71: caution is needed here. Orellana et al. discuss about micro and nanogels, determined with a different method with respect to the one reported here. When gels are present in the sea-surface microlayer, it will depend on their size distribution whether they will be part of the organic aerosol fraction or not. Aerosol particles smaller than 1 $\mu$m will be part of the accumulation mode of sea-spray aerosols, but when further aggregating and reaching sizes above 2.5$\mu$m they won't actually stay in the atmosphere longer than a few hours – as their size distribution is described as coarse mode aerosols. TEP as macromolecules are between accumulation and coarse mode but not ice-nucleating particles or cloud condensation nuclei. Another consideration is that if high wind speed are present (above 5 m/s), there might be increased

aggregation rates of TEP with solid particles which will favour the formation of nega-
tively buoyant aggregates that will sink out of the surface microlayer and surface waters
in general.

A: We agree with the reviewer that the microgels measured by Orellana et al. (2011),
defined as those stabilized with calcium bridges, may not fully correspond to TEP, de-
fined by their stainability with Alcian Blue (thus on their polysaccharide composition).
However, some studies have demonstrated that some TEP (about 30 %) are also sta-
bilized by divalent cations (Passow, 2002; Cisternas-Novoa et al., 2015). In addition,
even though TEP were measured in the particulate phase, we believe that TEP pre-
cursors could be measurable whenever TEP are present if they are in a dynamic equi-
librium with their precursors (Verdugo, 2012). Thus exopolymers in the dissolved and
colloidal phases, i.e. those potentially acting as CCN, would covary with TEP con-
centration (hypothesis yet to test). Furthermore, the exopolymer particles could de-
polymerise in the atmosphere due to ultraviolet light (Orellana and Verdugo, 2003) or
acidification (Chin et al., 1998) and form nano-sized particles (Karl et al., 2013). It is
also worth mentioning that Kuznetsova et al. (2005) found the presence of TEP (i.e.
Alcian Blue-stained polymers) in natural and simulated marine aerosols, and Russell et
al. (2010) showed the high carbohydrate composition of submicron aerosols in remote
regions of the North Atlantic and Arctic oceans that contained organic hydroxyl groups
from primary emissions of the ocean.

Since this is not the subject of the manuscript, we will not include this discussion but
will tone down a bit the statement referring to aerosol and clouds:

Line 70: "contributing to organic aerosol and possibly acting as cloud condensation
nuclei and ice nucleating particles (Orellana et al., 2011; Leck et al., 2013; Wilson et
al., 2015)."

R: Line 80: not just photolysis but also UV inhibited aggregation of precursor polymers
limits TEP formation.

A: We will add the following information in the revised version of the manuscript:

Line 101: "high solar radiation can stimulate TEP production by Prochlorococcus during cell decay (Iuculano et al., 2017), but also can limit TEP formation inhibiting the aggregation of the precursor polymers (Orellana and Verdugo, 2003)."

R: Line 102: What does this sentence mean? Please explain how HP affect TEP production and assembly of precursors.

A: Several experiments have found that the presence of bacteria stimulate or are necessary for TEP production by diatoms. Specifically, Guerrini et al. (1998) observed that the presence of bacteria during phosphate limitation conditions in batch cultures stimulated the production of polysaccharides by the diatom Cylindrotheca fusiformis. Gärdes et al. (2011) demonstrated that specific bacterial strains attached to the diatom Thalassiosira weissflogii was necessary for TEP production and suggested that direct interaction between bacteria and diatoms could be required for TEP formation. Moreover, through different mechanisms, HP seem to facilitate the self–assembly of dissolved precursors into TEP. In a seawater culture experiment, Sugimoto et al. (2007) observed that TEP formation appeared to be related with increases in bacterial abundance. Bacterial TEP production was not enough to explain the overall TEP formation and they suggested the self-assembly of TEP precursors coupled with bacterial growth. Ding et al. (2008) demonstrated that the amphiphilic exopolymers released by the bacterium Sagitula stellata induced DOM self-assembly and formation of marine microgels. We will add some of this previous information to better explain the processes involving prokaryote-TEP relationships.

Line 102: "HP have been found to stimulate TEP production by diatoms, suggesting that HP-diatom interaction is required for TEP formation (Guerrini et al., 1998; Gärdes et al., 2011),. HP may also facilitate TEP production from DOM self-assembly (Sugimoto et al., 2007), e.g., through the release of amphiphilic exopolymers that induce microgel formation ( Ding et al., 2008)."

R: Line 106: I suggest introducing the concept of biological carbon pump and the importance of TEP in ocean carbon cycle, as this is a central idea of the study. How much estimated primary production carbon is channeled into the TEP pool? (See Mari et al., 2017). This could also help making confrontations with phytoplankton-derived carbon, still estimates but could be interesting.

A: We will introduce the concept of biological carbon pump and the importance of TEP in the ocean carbon cycle. Beginning of the Introduction: "Due to their stickiness, TEP favour the formation of large aggregates of organic matter and organisms (typically named i.e. marine snow), enhancing particle ballast and sinking and thereby contributing to the biological carbon pump."

As for how much PP is channelled into TEP, we will add the following: "The aforementioned importance of TEP in carbon fluxes in the pelagic ocean can be further stressed by considering the following rough numbers: if the percentage of extracellular carbon release during planktonic primary production is generally constrained within 10-20 % (Nagata, 2000) but can reach >50% (López-Sandoval et al., 2011), and half of the extracellular release is in the form of reactive polysaccharides (Biddanda and Benner, 1997), then the production rate of TEP precursors may represent 5-10 %, but reach >25%, of planktonic primary production, without considering production by heterotrophs."

R: Lines 107-108: As mentioned already, TEP span over a wide range - DOC or POC is just an operational definition. From colloids (dissolved) to macrogels (particulate) (see Verdugo 2012 Annual Rev. of marine sciences).

A: We thank the reviewer for her/his comment. We will make the following changes to clarify it:

Line 107: "It is also important to determine the contribution of TEP as a constituent of the organic carbon pool to better understand its role in the organic matter cycling."

In the objectives section (end of introduction section), we will change the first sentence (line 110) to "we described the horizontal distribution of TEP (> 0.4 $\mu$m) in surface waters across a North–South transect in the Atlantic Ocean . . ."

Methods:

R: If you have DOC data, I think it would be worth showing them and looking for the missing fraction that drives POC underestimation with respect to TEP, as TEP are connecting both pools of organic matter. Can you provide a standard deviation or error estimation for POC filters?

A: Unfortunately, we don't have DOC data and there was only one replicate per POC measurement. However, we can add that the reproducibility of the elemental analyser used to measure POC (based on the coefficient of variation of the calibration slopes) is about 1 % for carbon. Regarding the coefficient of variation of the replicates, that takes into account the reproducibility of the whole process (sampling, filtering and analysis), we have obtained, in previous studies, a value of around 5 %.

We will add the following:

Line 153: "No POC replicates were run, but replication in a previous study yielded a coefficient of variation of around 5%."

R: Do you have wind speed information? This would be useful in estimating whether TEP could accumulate in the surface layer.

A: We have wind speed information but we can't estimate TEP relative accumulation in the surface layer as we only have data at one depth. The regression of TEP vs wind speed gave R2=0.2 in OAO and 0.3 in the SWAS, both with a negative slope. Contrasting results have been found in previous studies: Engel and Galgani (2016) found depletion of TEP in the SML above 5 m s-1, while earlier observations found enrichment in the microlayer also at higher wind speed (Wurl et al., 2009; Wurl et al., 2011).

Discussion:

R: Lines 317-319: Can you provide any reason why you think your values are higher than those observed in the Mediterranean Sea and Pacific Ocean? Is it related to nutrient concentration/time of year, different analysis method (e.g. spectroscopy vs microscopy for gel particles identification), depth?

A: We believe that one of the reasons is the depth. Mean TEP values in some of them (Ortega-Retuerta et al., 2010; Kodama et al., 2014; Ortega-Retuerta et al., 2017) correspond to the upper mixed layer depth or from 0 to 200 m. As TEP tend to accumulate in the surface and our values correspond only to the surface, this could explain the higher values obtained in our dataset. In fact, if we had provided integrated measurements within the photic layer, we would probably have obtained a lower mean TEP concentration. Another reason seems to be the different Chl a concentrations, as the main TEP producer is phytoplankton. Chl a concentration in the OAO ($0.4 \pm 0.2$ mg m-3 (0.2-0.6 mg m-3)) was generally higher than in the other studies referred in the Table. For example, in Iuculano et al. (2017) Chl a ranged 0.05-0.31 mg m-3, and in Kodama et al. (2014) Chl a averaged $0.05 \pm 0.01$ mg m-3. In some cases it is a pity that we don't have the average values, as the range could be a little bit misleading. However, in Ortega-Retuerta et al. (2010), TEP:Chl a ratio was higher than ours, suggesting that Chl a values were also low and gave rise to lower TEP. We can't forget either that differences in TEP chemical composition could cause differences in staining capacity. Regarding analytical methods, all the studies gathered in the table used the spectroscopic method, so this can't be the reason for the contrasting TEP concentrations.

We will briefly include these arguments in the discussion: Line 320: "Mean TEP values in some of them (Ortega-Retuerta et al., 2010; Kodama et al., 2014; Cisternas-Novoa et al., 2015; Ortega-Retuerta et al., 2017) correspond to the above mixed layer depth or from 0 to 100 or 200 m. As TEP tend to accumulate in the surface and our values correspond only to the surface (4 m), this could explain the higher values obtained in

our dataset. Another reason seems to be the different Chl a concentrations, as the main TEP producer is phytoplankton. Chl a concentration in the OAO (0.4 $\pm$ 0.2 mg m-3 (0.2-0.6 mg m-3)) was generally higher than in the other studies referred in the Table 2. For example, in Iuculano et al. (2017) Chl a ranged 0.05-0.31 mg m-3, and in Kodama et al. (2014) it averaged 0.05 $\pm$ 0.01 mg m-3. We can't discard either that differences in TEP chemical composition could cause differences in staining capacity."

R: Lines 355-356: The authors should mention here any limitation of the conversion factors.

A: We will mention it as follows:

Line 364: "Furthermore, conversion factors carry quite an uncertainty as pointed out in the Methods section".

R: Line 331: is the organic matter that influences HPA concentration and their TEP production or ..? How does the organic matter pool influences TEP formation? If you mean, by abiotic assembly of a pool of dissolved precursors, this concept should be mentioned early in the introduction.

A: We realize this sentence was ambiguous and we will change it. In the revised MS we now clarify the concept of abiotic formation. What we meant is that heterotrophic prokaryotes can be discharged directly with freshwater outflow, but also autochtonous microbes can be stimulated due to allochtonous DOM inputs. On the other hand, DOM inputs from freshwaters could also contain TEP and their precursors.

We will make the following change:

Line 328: "The nutrient–rich water in the region is responsible for the proliferation of phytoplankton and HP, which could partly explain the high TEP concentrations in this region. It is also known that large freshwater discharges occur in the shelf (Piola, 2005). These discharges could bring allochtonous HP directly to the shelf or bring DOM loads, which would stimulate autochtonous microbes. Besides, DOM inputs associated to

freshwater discharges could also contain TEP and their precursors."

R: Lines 370-374: Again, the fate of TEP depends on further aggregation processes. Generally less dense than water could accumulate in the surface microlayer but wind speeds, high heterotrophic activity, coagulation with other organic and mineral particles thanks to their stickiness should be mentioned to describe their fate in the area. Which one do you think would predominate?

A: We agree with the reviewer that TEP accumulation in the surface is the result of a complex suite of aggregation/consumption processes. Besides, the reference to the effects of TEP-richness on the fate of POC was a bit misplaced here, where we were discussing the potential reasons why TEP contribution to POC is larger in oligotrophic waters. We will remove the sentence to leave the paragraph:

Lines 367-374: "With our results taken all together, we hypothesize that in oligotrophic conditions TEP–C is the predominant POC fraction, because nutrient limitation favours TEP production by phytoplankton and limits TEP consumption by bacteria. Conversely, in eutrophic conditions, the predominant POC fraction depends on many variables like the community composition, the bloom stage, and sources of TEP different from phytoplankton."

R: Lines 409-410: Again on aerosol formation, it's a complex process and without any information on the size distribution of TEP in this study I would recommend caution in making such affirmations. Here, it's a bit a "stand alone" sentence without any further explanation, which does not make much sense. It should be expanded and explained better. Also, please see the paper by Quinn et al. 2014 "Contribution of sea surface carbon pool to organic matter enrichment in sea spray aerosol" Nature Geoscience, which actually breaks up the concept of organic aerosols related to phytoplankton blooms (and in this case, TEP).

A: We agree this sentence was a bit stand alone and too speculative, and have removed it.

R: Line 444: as mentioned, TEP can also be produced by aggregation of colloids in the absence of phytoplankton, that is, in the presence of polymeric precursors in the dissolved phase. Thus, it would be interesting to see the relationship of TEP to DOC or acidic sugars.

A: Unfortunately we don't have DOC or acidic sugars data to check this relationship. However, it is worth mentioning that covariation of TEP with DOC or dissolved carbohydrates are not always observed in the field (see for instance Ortega-Retuerta et al. (2009b) in the Southern Ocean). We will add the following information:

"TEP formation could have been enhanced by aggregation of colloids carried by freshwater discharges".

R: Line 454: UV also inhibits gel aggregation (Orellana and Verdugo 2003, Ultraviolet radiation blocks the organic carbon exchange between the dissolved phase and the gel phase in the ocean, Limnology and Oceanography). It should be mentioned here.

A: We will add this comment:

Line 454: "Our results suggest that the roles of UV radiation in breaking up TEP and/or limiting their formation from precursors overcome UV stress–induced TEP production."

REFERENCES

[revised manuscript text omitted]

Please also note the supplement to this comment:
https://www.biogeosciences-discuss.net/bg-2018-359/bg-2018-359-AC2-supplement.pdf

---

## Author Response (AR1)

**Reply to Referee#1**

*We thank the reviewer for his valuable comments on our manuscript. We answer below to each comment and question.*

**This is a good manuscript that provides excellent summary of TEP information. A good synthesis of data at hand despite the limitations of coverage in space (data collected only at 4m and at times the discussion is based on 1 sample to represent a hydrographic domain, say CU).**

*We thank the reviewer for his supportive comments. We were not trying to represent a hydrographic domain with a single sample, but we just treated the CU as an independent sample (i.e. removing it when calculating TEP averages and regression analyses between TEP and other environmental and biological parameters) due to its particularity, as indicated in the objectives section (end of the introduction). We are aware that some sentences could have given that impression and we did our best to fix this in the revised version of the MS. For example we have changed the following sentences:*

*Line 46: "with the maximum concentrations in the SWAS and in a station located at the edge of the Canary Coastal Upwelling (CU)"*

*Line 216: "and presented the minimum concentrations in the CU station and surroundings"*

*Lines 308-309: "namely in the station located in the CU and within the SWAS"*

*Line 366: "with the maximum value in the station located in the CU"*

*Lines 372-373: "The highest TEP:Chl a ratio of the entire transect observed in the station located in the CU was probably associated with the high relative abundance of diatoms and dinoflagellates."*

**The authors made the point that TEP contributes majorly to POC than phytos and HP based on the quantification of TEP, phytos and HP carbon pools estimated from available conversion factors. That the authors are well aware of limitations/approximations of these conversion factors, semi-quantitative nature estimations of TEP, phytos and HP pools (the last two are based on cell numbers) one would have expected the authors to critically evaluate their % contributions keeping the associated overall errors (methodology+conversion). This may not alter their conclusions but convinces the readers with appropriate comparisons having taken errors into account. I recommend minor revision of this manuscript before it is accepted for publication.**

*We added information in the manuscript regarding the errors associated with the methodology and conversion factors. More specifics are given in the responses below.*

**1. Lines 65-66: 'Enhancing particle sinking' – The authors may want to see open ocean TEP information from North Indian Ocean (Kumar et al., 1998)**

*We added the suggested reference in the revised version of the MS.*

**2. Line 67: 'can also ascent' gives a meaning that TEP float by themselves but these are mainly transported to surface microlayer by rising bubbles through scavenging**

> *We changed the sentence to (lines 70-72): "On their way to aggregation, and due to their low density, TEP and TEP–rich microaggregates formed near the surface may ascend and accumulate in the sea surface microlayer (SML) (Engel and Galgani, 2016), a process that is largely enhanced by bubble-associated scavenging (Azetsu-Scott and Passow, 2004; Wurl et al., 2009; Wurl et al., 2011b)."*

**3. Lines 109-110: "in situ studies of TEP distributions in the ocean are scarce, particularly in the open ocean (Table 2)". But Table 2 specifies TEP in surface layers. Kumar et al. (1998) and Ramaiah et al. (2000) provided the first TEP open ocean data from the Indian Ocean (see below for references).**

> *We thank the reviewer for drawing our attention to these references. We have specified in the text and figure legend that we are referring to surface measurements. Note that, for the sake of direct comparison with our study, Table 2 only listed TEP measurements conducted with the spectrophotometric method and Xanthan Gum calibration. However, in order to be more inclusive, we have added the indicated references, as follows.*

| Sargasso Sea | Oligotrophic | Spring, summer, autumn 2012 and spring 2013 | 0–100 | $21 \pm 2$– $57 \pm 3$ | 0.05–1 [c] | | Cisternas–Novoa et al. (2015) |
|---|---|---|---|---|---|---|---|
| North Indian Ocean -Arabian Sea -Bay of Bengal | -Eutrophic | -August 1996 -September 1996 | 0–1000 | -60 [j,k] (<5–102 [j]) -7–13 [c,j] | | | Kumar et al., (1998), Ramaiah et al., (2000) |

j: TEP concentrations were given in milligram equivalent of alginic acid $L^{-1}$ and absorbance was measured at 745 nm instead of 787 nm k: 0–50 m

**4. Line 115: 'entire POC' will also include non-living non-TEP organic carbon fraction. This was not addressed in the manuscript.**

> *We are aware that POC also includes other organic particle fractions such as non-living non-TEP organic carbon (for instance, cell fragments and proteinaceous - Coomassie stainable particles). In the present work we decided to compare our target variable, TEP, with the two pools of POC that are considered most abundant in sea water, namely phytoplankton and heterotrophic prokaryotes. We have added the following sentence in the results to clarify it (lines 265-271):*

*"To better explore the importance of TEP–C with respect to other major quantifiable POC pools, we estimated phytoplankton biomass (phyto–C) and HP biomass (HP–C) throughout the whole cruise (Fig. 2). It is worth mentioning that POC also includes other fractions of non-living non-TEP organic carbon (e.g., cell fragments and Coomassie stainable particles), but phytoplankton and heterotrophic prokaryotes are generally considered the most abundant in open sea water (Ortega-Retuerta et al., 2009b; Yamada et al., 2015). TEP–C contributed the most to the POC pool in the OAO, where it represented twice the share of phyto-C and HP-C. In the SWAS, conversely, TEP-C was not significantly different than phyto-C, and three times higher than HP-C (Fig. 3)."*

**5. Lines 147-148: Given the 18.7% difference in concentrations between TEP duplicates specify the errors in TEP-C estimation to compare with other org C-reservoirs.**

*Errors in TEP-C estimations averaged 8.4 µg C L$^{-1}$ (0.2- 70.3 µg C L$^{-1}$). However, rather than including these numbers, we have propagated the errors in the orf C reservoir calculations (error bars in Fig. 3).*

**6. Lines 175 to 202 and Lines 283-287: How accurate is the cell abundances counting of the respective biological groups? Please specify uncertainties involved. This is particularly important because each of subgroups will carry uncertainties in carbon per cell and that will be additive. Total uncertainties involved assume significance since a comparison is being made with TEP-C, where TEP estimation itself is semi-quantitative! For example, line 232-233 show phytoplankton biomass estimation carries nearly 50% of uncertainties in cell counts and cell C estimations! Authors discussion (Lines 343-353) on uncertainties in TEP-C contribution to POC arising from cell-C conversion and analytical artifacts is well appreciated. But the authors should help the readers by providing a comparative evaluation including errors in estimated carbon pools in a Table.**

*Replicates for the prokaryotic abundance measurement with flow cytometry were not done because the standard errors obtained are usually very low (i.e around 1.5 % in Pernice et al. (2015)).*

*Lines 190-191: "Only one replicate was analysed since standard errors of duplicates are usually very low (around 1.5 % in Pernice et al., 2015)."*

*Microscopic observations must be interpreted with caution due to the following (Kozlowski et al., 2011; Cassar et al., 2015):*
- *They are biased towards relatively large forms (> 5 µm) of phytoplankton groups with identifiable morphological characteristics*
- *Problems associated with biovolume estimates*
- *Problems with the microscopic identification of naked and small-celled groups*

*Lines 168-170: "Uncertainty sources for micro-phytoplankton biomass estimates are the conversion factors, biovolume estimates, and proper identification based on morphological characteristics, harder for naked cells*

*and those at the lower size edge (5-10 μm) (Kozlowski et al., 2011; Cassar et al., 2015)."*

*Regarding the phytoplankton biovolume-to-carbon conversion factors, we show the 95 % confidence intervals obtained by Menden-Deuer and Lessard (2000) for phytoplankton biomass estimation: log pg C cell$^{-1}$=log a (95 % C.I.) + b (95 % C.I.) x log V (μm$^3$), where log a is the y-intercept, b is the slope and 95% C.I. is the 95 % confidence intervals:*

*Protist plankton: log pg C cell$^{-1}$=log -0.665 (0.132) + 0.939 (0.041) x log V (μm$^3$)*
*Diatoms: log pg C cell$^{-1}$=log -0.541 (0.099) + 0.811 (0.028) x log V (μm$^3$)*

*Lines 164-167: "Cell C content was calculated using conversion equations of Menden-Deuer and Lessard (2000), log pg C cell-1 = log a (95 % confidence intervals) + b (95 % confidence intervals) × log volume (V; μm3): one for diatoms (log pg C cell-1 = log -0.541 (0.099) + 0.811 (0.028) × log V) and one for the other algae groups (log pg C cell-1 = log -0.665 (0.132) + 0.939 (0.041) × log V)."*

*As for the bacterial cell-to-carbon conversion factor, we added the following explanation:*

*Lines 194-197: "Ducklow (2000) summarized the carbon contents of free-living marine bacteria reported in the literature for a number of oceanic regions, bays and estuaries. The average ± standard deviation for open ocean regions was 12.3 ±2.5 fg C cell$^{-1}$. A factor of 12 fg C cell$^{-1}$ is equivalent to use the empirical equation proposed by Norland (1993), fgC cell$^{-1}$ = 0.12 (μm$^3$ cell vol)$^{0.72}$, for an average bacterial biovolume of 0.04 μm$^3$."*

*In relation to line 222-223: "The phytoplankton biomass was generally dominated by Prochlorococcus, with an average of 233 1.68 × 10$^5$ ±0.81 × 10$^5$ cells mL$^{-1}$, which corresponded to a biomass of 8.58 ± 4.16 μg C L$^{-1}$.", the standard deviation of biomass is not the uncertainty of the estimate, but the variability (standard deviation) of biomass along the Northeastern Subtropical Gyre.*

**7. Line 310: 'we present the first inventory of surface TEP concentration' – can the seawater samples collected from 4 m depth be treated as representative of surface layer to make an inventory? Here seems to be an incompatibility that needs to be clarified.**

*We agree with the reviewer that 4 m may at times not be representative of surface waters. Relatively high variability within the top surface meters has sometimes been observed (Wurl et al., 2009). However, 4 meters is usually considered as surface in most oceanographic studies, where sampling is mostly*

conducted either with the CTD rosette or with an underway pumping system. Nonetheless, the word 'inventory' may induce misunderstanding, and we changed it to 'distribution'. We have modified lines 290-293 of the manuscript as follows:

*"We present the first distribution of surface (4 m) TEP concentration along a latitudinal gradient in the Atlantic Ocean, covering both open sea and shelf waters. It is worth mentioning that vertical variability within the top surface meters (< 4 m) has sometimes been observed (Wurl et al., 2009), but 4 m is usually considered "surface ocean" in studies where samples are collected with either an oceanographic rosette or an underway pumping system."*

**8. Lines 360-364: Given the large uncertainties involved statements such as 'Only in one station of the SWAS phyto–C dominated the TEP–C (line 360-1)' and 'with the maximum concentrations in the edge of the Canary Coastal Upwelling (CU, n = 1) (lines 45-46)' may be avoided as these oversimplify a complex reality of spatial variability in horizontal and vertical (see line 310 comment above) dimensions.**

*As explained above, we were not trying to represent a hydrographic domain with a single sample, so, we made the appropriate changes in the text to clarify that we are just referring to our dataset without any purpose to generalise. E.g.:*

*Lines 45-46: "with the maximum concentrations in the station located in the edge of the Canary Coastal Upwelling (CU) and the SWAS"*

**10. Line 448: Please show the negative relation in a diagram.**

*We have added this plot as Fig. 4:*

[Figure]

*Figure 4: Relationship between the 24 hour–average (previous to sampling) solar irradiance (W m-2) and TEP (µg XG eq. L-1) in the OAO (CU sample excluded). The linear regression line is plotted and the equation indicated.*

**11. Lines 462-463: Figure 3 suggests that in spite of higher (nearly double) contribution of phytos to %POC in SWAS than in OAO, TEP and HP contributions to %POC are nearly the same. It appears that HP is more important in regulating TEP concentrations in the Atlantic, in general. This is slightly different from what has been said in lines 472-473 (The drivers of TEP distribution were primarily phytoplankton and, to a lesser extent, heterotrophic prokaryotes)**

*The identification of drivers of TEP distribution is based on regression analyses of covariation (Table 3). In OAO, the largest share of TEP variance is explained by Chl a ($R^2$=0.56) and phytoplankton biomass (0.47), particularly Synechococcus biomass (0.72), and in the SWAS it is phytoplankton biomass (0.62) followed by High nucleic acid containing prokaryotic heterotrophs (0.46). The fact that phytoplankton mainly drive TEP variability despite very different contribution to total POC is further exemplified by the large difference in the TEP:Chl a ratio between the two regions. In other words, the two regions are characterized by phytoplankton differently prone to TEP production, but in both phytoplankton are the main TEP drivers.*

**Reply to Referee #2**

**General comments**

**This is a good manuscript, well written and very informative about TEP distribution in surface waters of the Atlantic Ocean, taking into account many other studies. The main limitations, in my opinion, rely on the number of data collected and the depth, only at 4 m, which probably underestimates other processes related to TEP presence especially close to the depth of the chlorophyll maximum.**

> *We thank the reviewer for his/her positive general comments. We fully agree that it would be more informative to show vertical TEP profiles within the euphotic layer, but we consider that the study, which was carried out during a transit (i.e. no opportunities for CTD stations), adds valuable information for many processes happening at the ocean surface, as pointed out in the introduction.*

**As pointed out by the other referee, conversions factors may be approximative and a proper critical consideration on any limitation should be included in the manuscript.**

> *We have added information about the uncertainty of phytoplankton and heterotrophic prokaryotes biomass estimate (see responses to Referee#1).*

**Another general recommendation: TEP importance in processes such as air-sea gas exchange, aerosol formation, marine snow and carbon export and cycling should be better addressed in the whole study, as focal points of TEP influence on carbon dynamics, please see my comments in the introduction and discussion. I recommend that the following points are addressed before publication.**

**Abstract:**

**Lines 37-38: The authors should be aware that air-sea gas exchange and aerosol emissions are complex processes, which are not properly explained in the manuscript. I would thus remove this sentence that in the abstract appears a bit vague and would concentrate on the role of TEP in channeling the carbon produced by primary productivity (see Mari et al. 2017).**

> *We appreciate the reviewer's comment. We removed the sentence about aerosol emission. We have mentioned the role of TEP channeling the biological pump.*
>
> *Lines 36-39: "Transparent exopolymer particles (TEP) are a class of gel particles, produced mainly by microorganisms, which play important roles in biogeochemical processes such as carbon cycling and export. TEP (a) are colonized by carbon-consuming microbes; (b) mediate aggregation and sinking of organic matter and organisms, thereby contributing to the biological carbon pump; and (c) accumulate in the surface microlayer (SML) and affect air–sea gas exchange."*

**Lines 50-51: it could also be an inhibited TEP-aggregation by UV, not just breaking. I would rephrase the sentence.**

> *We have change this sentence in the revised version of the MS as follows (line 50:*
> *"suggesting that sunlight, particularly UV radiation, is more a sink than a source for TEP. "*
>
> *We also mentioned it in the introduction (see comments below), and the discussion:*
>
> *Lines 411-412: "Ultraviolet (UV) radiation causes TEP loss by photolysis (Ortega-Retuerta et al., 2009a) and inhibits TEP formation from precursors (Orellana and Verdugo, 2003)."*
>
> *Lines 414-415: "Our results suggest that the roles of UV radiation in breaking up TEP and/or limiting their formation from precursors overcome UV stress–induced TEP production."*

**Introduction:**

> *We thank the reviewer for his/her thorough effort to improve the introduction section of our manuscript.*

***Lines 63-75: The introduction is a bit vague, I would introduce the concept of a marine gel, the composition and cross-links in the molecule that make TEP water insoluble but still subject to fragmentation and further aggregation processes, and the size distribution in the ocean, mentioning the size range we are talking about. 0.4 µm falls into the truly dissolved phase, and a discussion on the continuum of sizes linking DOM and POM should be added.***

> *We have added the concept of DOM-POM continuum and evoke the gel polymer theory introducing a sentence like this in the revised version of the MS (lines 59-64): "Transparent exopolymer particles (TEP) are defined as a class of non–living organic particles in aqueous media, mainly consisting of acidic polysaccharides, which are stainable with Alcian Blue (Alldredge et al., 1993). They are formed from dissolved precursors that self–assemble to form TEP (operationally defined as particles > 0.4 µm) (Passow and Alldredge, 1994; Chin et al., 1998; Thuy et al., 2015). TEP are stabilized by covalent links or ionic strength (Cisternas-Novoa et al., 2015) and therefore, the formation and fragmentation of TEP from/to dissolved precursor material spans the dissolved to particulate continuum of organic matter in the sea.". However, one could consider 0.4 µm as a fraction included in the particulate phase if the 0.2 µm cutoff (one most widely used) is taken into account.*

**Several species can directly release TEP or macrogels, but such macromolecules can also form from dissolved abiotic material in the absence of phytoplankton (Chin, W.-C., Orellana, M.V., Verdugo, P., 1998. Spontaneous assembly of marine dissolved organic matter into polymer gels. Nature 391, 568–572.)**

*We already mentioned this process in former line 82 but we have made sure that the spontaneous assembly of DOM into TEP is not hidden (See comment above).*

**Moreover, the importance of TEP and marine snow should be mentioned. The role of TEP in the sea-surface microlayer should be either expanded or left out. The description presented here about air-sea gas exchange and aerosol is a bit vague and not precise. I would suggest spending more words on it, especially because 4 m depths is close to the surface so surface ocean processes and air-sea interaction should be properly mentioned. The role of TEP in the sea-surface microlayer depends on many factors: wind speed, primary productivity, and they are not the only class of gel particles present (e.g., highly productive region, see Engel and Galgani 2016, Biogeosciences, Wurl, O., Miller, L., Röttgers, R., and Vagle, S.: The distribution and fate of surface-active substances in the seasurface microlayer and water column, Mar. Chem., 115, 1–9, 2009. Wurl, O., Miller, L., and Vagle, S.: Production and fate of transparent exopolymer particles in the ocean, J. Geophys. Res., 116, C00H13, doi:10.1029/2011JC007342, 2011).**

*We added a few sentences to better introduce why TEP accumulates in the sea surface, implications and factors affecting this accumulation.*

*Lines 59-78: "Transparent exopolymer particles (TEP) are defined as a class of non–living organic particles in aqueous media, mainly consisting of acidic polysaccharides, which are stainable with Alcian Blue (Alldredge et al., 1993). They are formed from dissolved precursors that self–assemble to form TEP (operationally defined as particles > 0.4 μm) (Passow and Alldredge, 1994; Chin et al., 1998; Thuy et al., 2015). TEP are stabilized by covalent links or ionic strength (Cisternas-Novoa et al., 2015) and therefore, the formation and fragmentation of TEP from/to dissolved precursor material spans the dissolved to particulate continuum of organic matter in the sea. Due to their stickiness, TEP favour the formation of large aggregates of organic matter and organisms (typically named marine snow), enhancing particle ballast and sinking and thereby contributing to the biological carbon pump (Logan et al., 1995; Kumar et al., 1998; Passow et al., 2001; Burd and Jackson, 2009). The presence of TEP also affects the microbial food–web, as they can be used as a food source for zooplankton (Decho and Moriarty, 1990; Dilling et al., 1998; Ling and Alldredge, 2003) and heterotrophic prokaryotes (HP) (Passow, 2002b) through microbial colonization of aggregates (Alldredge et al., 1986; Grossart et al., 2006; Azam and Malfatti, 2007). On their way to aggregation, and due to their low density, TEP and TEP–rich microaggregates formed near the surface may ascend and accumulate in the sea surface microlayer (SML) (Engel and Galgani, 2016), a process that is largely enhanced by bubble–associated scavenging (Azetsu-Scott and Passow, 2004; Wurl et al., 2009; Wurl et al., 2011b). This accumulation in the SML, also contributed by local TEP production (Wurl et al., 2011b), can supress the air–sea exchange of $CO_2$ and other trace gases by acting as a physicochemical barrier or modifying sea surface hydrodynamics at low wind speeds (Calleja et al., 2008; Cunliffe et al., 2013; Wurl et al., 2016). Sea surface TEP can also be released to the atmosphere by bubble bursting (Zhou et al., 1998; Aller et al., 2005; Kuznetsova et al., 2005), contributing to organic aerosol and possibly acting as cloud*

*condensation nuclei and ice nucleating particles (Orellana et al., 2011; Leck et al., 2013; Wilson et al., 2015). All in all, TEP play important roles in microbial diversity, carbon cycling and carbon exports to both the deep ocean and the atmosphere."*

**Line 69: specify what do you mean by "affect air-sea gas exchange".**

*Some studies, revised in Cunliffe et al. (2013), show the influence of surface active components of the SML (including biogenic polysaccharides) on air-sea gas exchange, either acting as a physicochemical barrier or modifying sea surface hydrodynamics, which in turn results in a suppression of air-water gas exchange. For example, Calleja et al. (2008) found that the organic matter content of the surface water supressed $CO_2$ gas exchange between the air and the ocean at low and intermediate wind speeds (> 5 m s$^{-1}$). Wurl et al. (2016) found enrichments of TEP, POC, PON, total prokaryotic cell numbers and picophytoplankton abundances in sea microlayers at multiple stations of different regions, compared to the underlying bulk water, being higher in slick surfaces than non-slick ones, and estimated that slicks could reduce $CO_2$ fluxes by up to 15 %, which highlight the importance of slicks in regulating air-sea interactions. Jenkinson et al. (2018) reviewed recently known and suspected mechanical aspects of how biologically produced organic matter modulates air-sea fluxes of $CO_2$.*

*We have briefly added some of this information in the introduction section. See comment above.*

**Lines 70-71: caution is needed here. Orellana et al. discuss about micro and nanogels, determined with a different method with respect to the one reported here. When gels are present in the sea-surface microlayer, it will depend on their size distribution whether they will be part of the organic aerosol fraction or not. Aerosol particles smaller than 1 μm will be part of the accumulation mode of sea-spray aerosols, but when further aggregating and reaching sizes above 2.5μm they won't actually stay in the atmosphere longer than a few hours – as their size distribution is described as coarse mode aerosols. TEP as macromolecules are between accumulation and coarse mode but not ice-nucleating particles or cloud condensation nuclei. Another consideration is that if high wind speed are present (above 5 m/s), there might be increased aggregation rates of TEP with solid particles which will favour the formation of negatively buoyant aggregates that will sink out of the surface microlayer and surface waters in general.**

*We agree with the reviewer that the microgels measured by Orellana et al. (2011), defined as those stabilized with calcium bridges, may not fully correspond to TEP, defined by their stainability with Alcian Blue (thus on their polysaccharide composition). However, some studies have demonstrated that some TEP (about 30 %) are also stabilized by divalent cations (Passow, 2002; Cisternas-Novoa et al., 2015). In addition, even though TEP were measured in the particulate phase, we believe that TEP precursors could be measurable whenever TEP are present if they are in a dynamic equilibrium with their precursors (Verdugo, 2012). Thus exopolymers in the dissolved and colloidal phases, i.e. those potentially acting as CCN, would covary with TEP*

*concentration (hypothesis yet to test). Furthermore, the exopolymer particles could depolymerise in the atmosphere due to ultraviolet light (Orellana and Verdugo, 2003) or acidification (Chin et al., 1998) and form nano-sized particles (Karl et al., 2013). It is also worth mentioning that Kuznetsova et al. (2005) found the presence of TEP (i.e. Alcian Blue-stained polymers) in natural and simulated marine aerosols, and Russell et al. (2010) showed the high carbohydrate composition of submicron aerosols in remote regions of the North Atlantic and Arctic oceans that contained organic hydroxyl groups from primary emissions of the ocean.*

*Since this is not the subject of the manuscript, we have not included this discussion but have toned down a bit the statement referring to aerosol and clouds:*

*Lines 76-77: "contributing to organic aerosol and possibly acting as cloud condensation nuclei and ice nucleating particles (Orellana et al., 2011; Leck et al., 2013; Wilson et al., 2015)."*

**Line 80: not just photolysis but also UV inhibited aggregation of precursor polymers limits TEP formation.**

*We have added the following information in the revised version of the manuscript:*

*Lines 98-99: "high solar radiation can stimulate TEP production by Prochlorococcus during cell decay (Iuculano et al., 2017), but also can limit TEP formation inhibiting the aggregation of the precursor polymers (Orellana and Verdugo, 2003)."*

**Line 102: What does this sentence mean? Please explain how HP affect TEP production and assembly of precursors.**

*Several experiments have found that the presence of bacteria stimulate or are necessary for TEP production by diatoms. Specifically, Guerrini et al. (1998) observed that the presence of bacteria during phosphate limitation conditions in batch cultures stimulated the production of polysaccharides by the diatom Cylindrotheca fusiformis. Gärdes et al. (2011) demonstrated that specific bacterial strains attached to the diatom Thalassiosira weissflogii was necessary for TEP production and suggested that direct interaction between bacteria and diatoms could be required for TEP formation.*
*Moreover, through different mechanisms, HP seem to facilitate the self–assembly of dissolved precursors into TEP. In a seawater culture experiment, Sugimoto et al. (2007) observed that TEP formation appeared to be related with increases in bacterial abundance. Bacterial TEP production was not enough to explain the overall TEP formation and they suggested the self-assembly of TEP precursors coupled with bacterial growth. Ding et al. (2008) demonstrated that the amphiphilic exopolymers released by the bacterium Sagitula stellata induced DOM self-assembly and formation of marine microgels.*
*We have added some of this previous information to better explain the processes involving prokaryote-TEP relationships.*

*Lines 100-103: "HP have been found to stimulate TEP production by diatoms, suggesting that HP-diatom interaction is required for TEP formation (Guerrini et al., 1998; Gärdes et al., 2011). HP may also facilitate TEP production from DOM self-assembly (Sugimoto et al., 2007), e.g., through the release of amphiphilic exopolymers that induce microgel formation ( Ding et al., 2008)."*

**Line 106: I suggest introducing the concept of biological carbon pump and the importance of TEP in ocean carbon cycle, as this is a central idea of the study. How much estimated primary production carbon is channeled into the TEP pool? (See Mari et al., 2017). This could also help making confrontations with phytoplankton-derived carbon, still estimates but could be interesting.**

*We have introduced the concept of biological carbon pump and the importance of TEP in the ocean carbon cycle. Beginning of the Introduction (lines 64-67): "Due to their stickiness, TEP favour the formation of large aggregates of organic matter and organisms (typically named marine snow), enhancing particle ballast and sinking and thereby contributing to the biological carbon pump (Logan et al., 1995; Kumar et al., 1998; Passow et al., 2001; Burd and Jackson, 2009)."*

*As for how much PP is channelled into TEP, we added the following: Lines 104-109: "The aforementioned importance of TEP in carbon fluxes in the pelagic ocean can be further stressed by considering the following rough numbers: if the percentage of extracellular carbon release during planktonic primary production is generally constrained within 10-20 % (Nagata, 2000) but can reach >50% (López-Sandoval et al., 2011), and half of the extracellular release is in the form of reactive polysaccharides (Biddanda and Benner, 1997), then the production rate of TEP precursors may represent 5-10 %, but reach >25%, of planktonic primary production, without considering production by heterotrophs."*

**Lines 107-108: As mentioned already, TEP span over a wide range - DOC or POC is just an operational definition. From colloids (dissolved) to macrogels (particulate) (see Verdugo 2012 Annual Rev. of marine sciences).**

*We thank the reviewer for her/his comment. We made the following changes to clarify it:*

*Lines 109-110: "This calls for the need to quantify their occurrence across the oceans, elucidate their main distribution drivers, and determine their contribution to the organic carbon reservoir."*

*In the objectives section (end of introduction section), we changed the first sentence (lines 111-112) to "we describe the horizontal distribution of TEP (> 0.4 μm) in surface waters across a North–South transect in the Atlantic Ocean,"*

**Methods:**

**If you have DOC data, I think it would be worth showing them and looking for the missing fraction that drives POC underestimation with respect to TEP, as TEP are connecting both pools of organic matter. Can you provide a standard deviation or error estimation for POC filters?**

> *Unfortunately, we don't have DOC data and there was only one replicate per POC measurement. However, we added that the reproducibility of the elemental analyser used to measure POC (based on the coefficient of variation of the calibration slopes) is about 1 % for carbon. Regarding the coefficient of variation of the replicates, which takes into account the reproducibility of the whole process (sampling, filtering and analysis), we have obtained, in previous studies, a value of around 5 %.*
>
> *We added the following:*
>
> *Line 147-148: "No POC replicates were run, but replication in a previous study yielded a coefficient of variation of around 5 %."*

**Do you have wind speed information? This would be useful in estimating whether TEP could accumulate in the surface layer.**

> *We have wind speed information but we can't estimate TEP relative accumulation in the surface layer as we only have data at one depth. The regression of TEP vs wind speed gave $R^2=0.2$ in OAO and 0.3 in the SWAS, both with a negative slope. Contrasting results have been found in previous studies: Engel and Galgani (2016) found depletion of TEP in the SML above 5 m s$^{-1}$, while earlier observations found enrichment in the microlayer also at higher wind speed (Wurl et al., 2009; Wurl et al., 2011).*

**Discussion:**

**Lines 317-319: Can you provide any reason why you think your values are higher than those observed in the Mediterranean Sea and Pacific Ocean? Is it related to nutrient concentration/time of year, different analysis method (e.g. spectroscopy vs microscopy for gel particles identification), depth?**

> *We believe that one of the reasons is the depth. Mean TEP values in some of them (Ortega-Retuerta et al., 2010; Kodama et al., 2014; Ortega-Retuerta et al., 2017) correspond to the upper mixed layer depth or from 0 to 200 m. As TEP tend to accumulate in the surface and our values correspond only to the surface, this could explain the higher values obtained in our dataset. In fact, if we had provided integrated measurements within the photic layer, we would probably have obtained a lower mean TEP concentration.*
> *Another reason seems to be the different Chl a concentrations, as the main TEP producer is phytoplankton. Chl a concentration in the OAO (0.4 ± 0.2 mg m$^{-3}$ (0.2-0.6 mg m$^{-3}$)) was generally higher than in the other studies referred in the Table. For example, in Iuculano et al. (2017) Chl a ranged 0.05-0.31 mg m$^{-3}$, and in Kodama et al. (2014) Chl a averaged 0.05 ± 0.01 mg m$^{-3}$. In some cases it*

*is a pity that we don't have the average values, as the range could be a little bit misleading. However, in Ortega-Retuerta et al. (2010), TEP:Chl a ratio was higher than ours, suggesting that Chl a values were also low and gave rise to lower TEP. We can't forget either that differences in TEP chemical composition could cause differences in staining capacity. Regarding analytical methods, all the studies gathered in the table used the spectroscopic method, so this can't be the reason for the contrasting TEP concentrations.*

*We have briefly included these arguments in the discussion:*

*Lines 300-307: "Mean TEP values in some of them (Ortega-Retuerta et al., 2010; Kodama et al., 2014; Cisternas-Novoa et al., 2015; Ortega-Retuerta et al., 2017) correspond to the above mixed layer depth or from 0 to 100 or 200 m. As TEP tend to accumulate in the surface and our values correspond only to the surface (4 m), this could explain the higher values obtained in our dataset. Another reason seems to be the different Chl a concentrations, as the main TEP producer is phytoplankton. Chl a concentration in the OAO (0.4 ± 0.2 mg m$^{-3}$ (0.2-0.6 mg m$^{-3}$)) was generally higher than in the other studies referred in the Table 2. For example, in Iuculano et al. (2017) Chl a ranged 0.05-0.31 mg m$^{-3}$, and in Kodama et al. (2014) it averaged 0.05 ± 0.01 mg m$^{-3}$. We can't discard either that differences in TEP chemical composition could cause differences in staining capacity."*

**Lines 355-356: The authors should mention here any limitation of the conversion factors.**

*We have mentioned it as follows:*

*Line 343-344: "Furthermore, conversion factors carry quite an uncertainty as pointed out in the Methods section".*

**Line 331: is the organic matter that influences HPA concentration and their TEP production or ..? How does the organic matter pool influences TEP formation? If you mean, by abiotic assembly of a pool of dissolved precursors, this concept should be mentioned early in the introduction.**

*We realized this sentence was ambiguous and changed it. In the revised MS we now clarify the concept of abiotic formation. What we meant is that heterotrophic prokaryotes can be discharged directly with freshwater outflow, but also autochtonous microbes can be stimulated due to allochtonous DOM inputs. On the other hand, DOM inputs from freshwaters could also contain TEP and their precursors.*

*We have made the following change:*

*Lines 313-317: "The nutrient–rich water in the region is responsible for the proliferation of phytoplankton and HP, which could partly explain the high TEP*

*concentrations in this region. It is also known that large freshwater discharges occur in the shelf (Piola, 2005). These discharges could bring allochtonous HP directly to the shelf or bring DOM loads, which would stimulate autochtonous microbes. Besides, DOM inputs associated to freshwater discharges could also contain TEP and their precursors."*

**Lines 370-374: Again, the fate of TEP depends on further aggregation processes. Generally less dense than water could accumulate in the surface microlayer but wind speeds, high heterotrophic activity, coagulation with other organic and mineral particles thanks to their stickiness should be mentioned to describe their fate in the area. Which one do you think would predominate?**

*We agree with the reviewer that TEP accumulation in the surface is the result of a complex suite of aggregation/consumption processes. Besides, the reference to the effects of TEP-richness on the fate of POC was a bit misplaced here, where we were discussing the potential reasons why TEP contribution to POC is larger in oligotrophic waters. We have removed the sentence to leave the paragraph:*

*Lines 347-351: "With our results taken all together, we hypothesize that in oligotrophic conditions TEP–C is the predominant POC fraction, because nutrient limitation favours TEP production by phytoplankton and limits TEP consumption by bacteria. Conversely, in eutrophic conditions, the predominant POC fraction depends on many variables like the community composition, the bloom stage, and sources of TEP different from phytoplankton."*

**Lines 409-410: Again on aerosol formation, it's a complex process and without any information on the size distribution of TEP in this study I would recommend caution in making such affirmations. Here, it's a bit a "stand alone" sentence without any further explanation, which does not make much sense. It should be expanded and explained better. Also, please see the paper by Quinn et al. 2014 "Contribution of sea surface carbon pool to organic matter enrichment in sea spray aerosol" Nature Geoscience, which actually breaks up the concept of organic aerosols related to phytoplankton blooms (and in this case, TEP).**

*We agree this sentence was a bit stand alone and too speculative, and have removed it.*

**Line 444: as mentioned, TEP can also be produced by aggregation of colloids in the absence of phytoplankton, that is, in the presence of polymeric precursors in the dissolved phase. Thus, it would be interesting to see the relationship of TEP to DOC or acidic sugars.**

*Unfortunately we don't have DOC or acidic sugars data to check this relationship. However, it is worth mentioning that covariation of TEP with DOC or dissolved carbohydrates are not always observed in the field (see for instance Ortega-Retuerta et al. (2009b) in the Southern Ocean). We have added the following information (lines 407-408):*

*"Moreover, as mentioned before, in these shelf waters TEP formation could have been further modulated by aggregation of colloids carried by freshwater discharges".*

**Line 454: UV also inhibits gel aggregation (Orellana and Verdugo 2003, Ultraviolet radiation blocks the organic carbon exchange between the dissolved phase and the gel phase in the ocean, Limnology and Oceanography). It should be mentioned here.**

*We added this comment:*

*Lines 414-415: "Our results suggest that the roles of UV radiation in breaking up TEP and/or limiting their formation from precursors overcome UV stress–induced TEP production."*

**REFERENCES**

[revised manuscript text omitted]